# Asymmetry, Not Capability: Evaluation Shapes Tool-Augmented PII Detection in Small Language Models

## Abstract

Agent augmentation is widely assumed to help, yet for small language models, its measured benefit on PII detection depends more on how detection is scored than on model capability. A controlled ablation spans four open-weight models (Gemma 2 9B, Llama 3.1 8B, Mistral 7B, Qwen 2.5 7B) and four conditions (zero-shot, +Docs, +Tool, +Skills) on a stratified 2,000-sample benchmark, scored under one symmetric label-alignment map and two co-primary metrics. At the *type* level, zero-shot beats both tool-augmented conditions for every model by 6–17 F1 points ($p < 0.0001$), and on average few-shot and chain-of-thought prompting match or trail it, ruling out a weak baseline. At the *span* level, the ranking inverts. That inversion does not reflect better detection. The models identify PII types at essentially the same rate as the detector-backed conditions (27.2% versus 27.0% of span-bearing gold entities), but they cannot indicate their locations, yielding an exact-boundary F1 of 0.001. This is because the detector supplies character offsets programmatically, whereas the model must generate them, which these models effectively could not do. The span-level "tool advantage" measures offset provenance, not capability. Much of the type-level gap is a label-alignment artifact, and the residual is an interface effect, since the recall-tuned detector over-proposes candidates that a precision-weighted metric penalizes. The inversion holds when either consequential label bridge is removed, or both. Both artifacts are asymmetries between what a model emits and what a tool emits, and both are structural properties of any model-plus-detector pipeline. The pattern persists without quantization and one model generation up (Qwen 2.5 14B). Such asymmetries, not model capability, largely determine whether agentic scaffolding appears to help.

## 1 Introduction

The proliferation of agentic AI systems has introduced a new class of prompting strategies in which language models are given access to external tools, domain documentation, and structured skill definitions at inference time (Yao et al., 2023; Schick et al., 2023; Chase, 2022). If a model lacks reliable internal representations of a specialized domain, injecting authoritative external references should improve task performance. This assumption underpins a broad class of retrieval-augmented generation (RAG) systems (Lewis et al., 2020), tool-use frameworks (Chase, 2022), and the emerging concept of *Agent Skills* (modular, reusable context documents that encode procedural knowledge, tool syntax, and domain constraints for a specific task).

PII detection is a natural testbed for this hypothesis. It is a high-stakes structured extraction task with a well-defined label schema, and it is increasingly delegated to SLMs in agentic data pipelines where latency, cost, and privacy constraints preclude the use of frontier-scale models (Lison et al., 2021; Lukas et al., 2023). Sending sensitive text to API-based frontier models raises privacy and security concerns, so local deployment is often preferred for these workloads (Sun et al., 2025). The domain also has a mature programmatic library ecosystem including Microsoft Presidio (Mendels et al., 2018), spaCy NER (Honnibal et al., 2020), and PII-Codex (Rosado, 2023), which provide structured canonical taxonomies against which model predictions can be systematically compared, though programmatic detection accuracy varies by entity type. This yields a concrete, falsifiable hypothesis for why an agentic formulation might help even on a task whose answer lies in the input text. Programmatic detectors encode reliable rules for structured identifiers (date, social-security,

and IP formats), so tool access should raise recall on exactly those types. The empirical question is whether that type-specific benefit persists under aggregation across the full schema, or is outweighed by the cost of deferring to a rule-based tool on the many types where a model's contextual understanding is stronger.

A four-condition ablation was designed to isolate the effects of documentation, tool access, and skills injection on PII detection accuracy in four open-weight SLMs (7–9B parameters), including Gemma 2 9b (Gemma Team, 2024), Llama 3.1 8b (Grattafiori et al., 2024), Mistral 7b (Jiang et al., 2023), and Qwen 2.5 7b (Yang et al., 2024).

The conditions are:

1. *Zero-shot (+ZS):* The model receives a task prompt and sample text with no external augmentation.

2. *Documentation (+Docs):* The model additionally receives the PII-Codex reference documentation in context (single-turn; no tool execution).

3. *Tool-augmented (+Tool):* The model is given access to the `analyze_pii` tool (PII-Codex execution) via a LangGraph-based SkillsAgent, and may call it before returning a final answer.

4. *Skills-augmented (+Skills):* In addition to tool access, the model may discover and read a structured PII detection Skill document (`SKILL.md`) via `list_skills` and `view_skill` tool calls before calling `analyze_pii`.

Beyond these four conditions, the design adds three baselines and two robustness controls. The baselines (PII-Codex run standalone, isolating the library; few-shot prompting; and chain-of-thought prompting) guard against a weak-comparison artifact. Averaged across the four models, neither parametric baseline exceeds zero-shot (few-shot 0.628 and chain-of-thought 0.606 against zero-shot 0.631), though per-model effects are small and mixed; the standalone library trails every model condition. The zero-shot comparison is therefore not a weak baseline. Two controls extend the design beyond the 7–9B class. The first re-runs the ablation on a larger model (Qwen2.5-14B), and the second re-runs it unquantized (fp16/bf16), each testing whether an observed effect is bound to model scale or quantization rather than to the evaluation pipeline.

Using a stratified benchmark of 2,000 PII-annotated English samples from three public datasets, zero-shot prompting outperforms both tool-augmented conditions for all four models at the entity-type level, contrary to common assumptions. However, at the span level the ranking inverts, not because the detector localizes better, but because it returns character offsets programmatically while the model must generate them and effectively cannot (Section 4.2). Two main causes contribute to the type-level degradation. The first is a label-alignment artifact that can be balanced through symmetric mapping, and the second is a model-library interface issue in which the rule-based detector over-predicts and lacks context. This pattern remains consistent without quantization and persists even with a single 14B model (Qwen2.5-14B).

The tool condition compares *non-fine-tuned* tool calling against zero-shot extraction, and once invoked, the tool's output (not the model) determines the canonical predictions. Wrapping a non-fine-tuned 7–14B model around a rule-based PII library does not help on this static task under fair evaluation, and the augmented conditions do not recover more PII than the model does on its own. This is not a claim that parametric knowledge universally beats tools. The methodological claim is not scoped in the same way, since the asymmetries that produce this result are structural to any model-plus-detector pipeline. The study therefore carries a measurement lesson alongside its empirical one. In mixed-mode pipelines, what an evaluation measures, and not what a system can do, often decides whether an intervention appears beneficial.

This aligns with the view that modern language models already possess strong inherent reasoning and text-understanding capabilities (Yu et al., 2024) and need not rely on external tools for a static extraction task such as PII detection. It also aligns with recent evidence that arbitrary tool access can actively degrade performance compared to zero-shot, when retrieval errors and scaffolding overhead outweigh tool benefits (Luo et al., 2025; Qian et al., 2025; Xu et al., 2025). LLMs struggle to decide whether to use a tool (Ning et al., 2024); zero-shot prompting avoids that pitfall by not deferring to a tool that does not improve detection here. The finding has direct implications for how agentic scaffolding should be evaluated and deployed in production NLP pipelines.

## 2 Related Work

**PII detection with language models.** Named entity recognition (NER) approaches to PII detection have a long history in NLP (Lample et al., 2016; Devlin et al., 2019). More recently, instruction-tuned LLMs have been evaluated as zero-shot or few-shot entity extractors, though their out-of-the-box performance on strict sequence-labeling tasks often lags behind supervised baselines without highly specialized prompt adaptations (Wang et al., 2025). Separate work demonstrates that language models are highly susceptible to leaking sensitive PII depending on the prompt design and the entity type targeted by an attacker, underscoring the critical need for reliable, external PII detection safeguards in agentic pipelines (Lukas et al., 2023). PII-Codex (Rosado, 2023) builds on Microsoft Presidio (Mendels et al., 2018) and draws on the information-sensitivity typology (Milne et al., 2016) and the risk identification continuum (Schwartz & Solove, 2011). It supplies a unified taxonomy and a programmatic evaluation framework for type-level scoring against a canonical schema.

**Tool use and RAG in language models.** Surveys of natural language reasoning suggest that modern language models already possess strong inherent reasoning and text-understanding capabilities (Yu et al., 2024); for a static, pattern-recognition task like PII detection, external tools may therefore be redundant or harmful. Conversely, tools are necessary when models must incorporate real-time, rapidly changing, or newly updated information that cannot be captured by static pretrained weights (Yu & Ji, 2024); PII detection, by contrast, is a static extraction task on which the models studied here are not improved by tool access, and forcing an agent to invoke tools adds overhead without a corresponding gain.

ReAct (Yao et al., 2023) and Toolformer (Schick et al., 2023) established that language models can interleave reasoning and tool execution. Subsequent work has shown that tool use can substantially improve performance on knowledge-intensive tasks, though gains are not universal and depend heavily on instruction-following fidelity and the schema alignment between tool outputs and the evaluation metric (Qin et al., 2024). Providing models with arbitrary tool access can actively degrade performance compared to zero-shot, as error rates from unnecessary retrieval, inappropriate invocation, and scaffolding often outweigh tool benefits (Luo et al., 2025; Qian et al., 2025; Xu et al., 2025). LLM-generated internal knowledge can surpass externally retrieved knowledge. External retrieval often introduces irrelevant or poorly coherent information that harms downstream performance more than minor factual errors in parametric memory (Chen et al., 2023). For many tasks, parametric knowledge is sufficient and "tool overuse" lowers accuracy (Luo et al., 2025; Qian et al., 2025). SLMs generally fail to execute multi-turn tool-discovery loops out-of-the-box (Patil et al., 2024; Qin et al., 2024). Work arguing that SLMs are well suited to agentic roles nonetheless makes the same prescription, holding that they become reliable tool callers once fine-tuned for a specialized role (Belcak et al., 2025; Jhandi et al., 2025). Toolformer showed that a small model can excel at tool use when fine-tuned to predict tool calls (Schick et al., 2023); out-of-the-box scaffolding without such fine-tuning is a poor fit for the 7–9B class (Patil et al., 2024). The study contributes an evaluation of tool use in the specific context of structured information extraction in the 7–9B parameter class, with a robustness check one model generation up. It also contributes a methodological result that is not tied to that setting. Mixed model-tool evaluation is distorted by asymmetries between what a model emits and what a tool emits, in label ontology and in the provenance of character offsets, and these asymmetries, left unreconciled, manufacture capability differences that do not exist.

**Agent Skills and skill injection.** Injecting modular "skill" documents (reusable context that encodes procedural knowledge, tool syntax, and domain constraints) into a model's context at inference time is a practitioner pattern that has so far received limited formal evaluation. It should be distinguished from a separate and more developed line of work on *skill acquisition and optimization*, in which an agent learns, composes, or iteratively refines a library of skills against task feedback rather than reading a fixed document at inference (Yang et al., 2026; Alzubi et al., 2026). Those approaches treat skills as learned, evolving artifacts; the inference-time injection studied here instead supplies a static, hand-written document and asks whether merely placing it in context helps a frozen model. The value of such a combination is, moreover, task-dependent: PokerSkill (Li et al., 2026) reports that pairing expert-authored rule-based skills with an LLM beats either component alone on strategic poker, whereas for static PII extraction the model is strongest

unaugmented at the type level, though the decomposition here shows the two recover PII at parity and differ mainly in the form of what they emit, underscoring that whether skill or tool grounding helps depends on the task and the quality of the underlying library. To the best of the author's knowledge, this is the first controlled ablation of inference-time skill-document injection for PII detection, with pre-specified conditions and post-hoc statistical testing.

## 3 Methods

### 3.1 Benchmark compilation

A stratified benchmark of 2,000 English-language PII-annotated samples was compiled from three public HuggingFace datasets, AI4Privacy (Ai4Privacy, 2024) (`ai4privacy/pii-masking-300k`), NVIDIA Nemotron-PII (Steier et al., 2025) (`nvidia/Nemotron-PII`), and Gretel PII Masking (Gretel AI, 2024) (`gretelai/gretel-pii-masking-en-v1`).

Let $D = \{(x_i, G_i)\}_{i=1}^{N}$ with $N = 2,000$ denote the benchmark, where $x_i$ is a text sample and $G_i = \{(s_{ij}, y_{ij})\}$ is the set of ground-truth entity spans and types for sample $i$. Source distributions in the final benchmark are shown in Table 1. The mix of PII types varies by source; type-level counts are in Appendix Table 10.

Table 1: Benchmark source distribution ($N = 2,000$). Samples and per-source entity counts as recorded during stratification. Total entity span count for the compiled benchmark is given in Appendix Table 10.

| Source | Samples | Entity spans (source) | Mean/sample |
|---|---|---|---|
| AI4Privacy | 1,132 | 7,112 | 6.28 |
| Gretel PII Masking | 581 | 2,416 | 4.16 |
| NVIDIA Nemotron-PII | 287 | 1,842 | 6.42 |
| **Total** | **2,000** | **11,370** | **5.69** |

**Language and locale filtering.** AI4Privacy was filtered to rows with `language == "English"`; NVIDIA Nemotron-PII to `locale == "us"`; Gretel requires no filter (English-only dataset). Results apply to English text and US-locale PII types only and do not generalize to other locales or languages.

**Label mapping and exclusion.** Source labels were normalized and mapped to PII-Codex (Rosado, 2023) canonical types (`PIIType.name`) via a fixed 136-entry mapping table. Records containing any PII instance with no PII-Codex mapping were excluded, so all ground-truth labels in the benchmark are fully mappable. The final set covers 21 PII types. Ground-truth entity span counts (each contiguous span of a given type counted once) by type for the full benchmark are in Appendix Table 10; the most frequent types are `PERSON`, `DATE_TIME`, `LOCATION`, `ADDRESS`, and `EMAIL_ADDRESS` (Table 2).

### 3.2 Models

Four open-weight instruction-tuned models were evaluated: Gemma 2 9B, Llama 3.1 8B, Mistral 7B, and Qwen 2.5 7B. The main study used 4-bit quantized MLX variants (`mlx-community`) on a 2020 MacBook Pro M1 Pro Max (64 GB RAM). Two robustness controls add further runs. The first is an unquantized (fp16/bf16) variant of each model on a stratified 300-sample subset. The second is a larger model (Qwen2.5-14B, 4-bit) on the full benchmark (Section 5.6, Appendix F). Prompts and chat templates were identical across models and conditions; seed 42 was fixed for all runs.

### 3.3 Conditions

The study comprises four primary conditions, three additional baselines, and two robustness controls. The four primary conditions are evaluated on all 2,000 benchmark samples across the four models, yielding

Table 2: Selected frequent PII types by ground-truth entity count (main study benchmark, descending). Full distribution in Appendix Table 10.

| PII Type | Entity count |
|---|---|
| PERSON | 2,324 |
| DATE_TIME | 1,572 |
| LOCATION | 1,068 |
| ADDRESS | 862 |
| EMAIL_ADDRESS | 632 |
| US_SOCIAL_SECURITY_NUMBER | 632 |
| PHONE_NUMBER | 566 |
| DATE | 531 |
| IP_ADDRESS | 515 |
| US_DRIVERS_LICENSE_NUMBER | 383 |
| HEALTH_INSURANCE_ID | 343 |

32,000 prediction rows (2,000 samples $\times$ 4 models $\times$ 4 conditions); the baselines (Detector-only, Few-shot, and Chain-of-thought, described below) and the robustness controls (a 14B model and an unquantized re-run, Section 5.6) extend this core ablation. Each condition $c$ defines an inference pipeline $f_c \colon \mathcal{X} \to \mathcal{P}(\mathcal{S} \times \mathcal{Y}_c)$ mapping input text $x_i$ to a predicted set of entity spans and labels $\hat{G}_{i,c} = f_c(x_i)$. The four pipelines are shown in Figure 1.

**Zero-shot (ZS).** Single-turn prompt containing the task instruction and sample text. No external augmentation. The model returns a JSON array of PII predictions.

**Documentation (+Docs).** Same single-turn path with the PII-Codex reference documentation prepended to the prompt. No tool execution.

**Tool-augmented (+Tool).** The model is given access to `analyze_pii`, which executes PII-Codex on the sample text, via a custom SkillsAgent created with LangGraph (LangChain, 2024; Chase, 2022). Tool-call detection uses strict bracket syntax (`[TOOL_CALL: analyze_pii]`) with a lenient intent-based fallback. If no tool call is detected, the model's first response is taken as the final answer. Once the tool fires ($\sim$99.99% of runs; Section 4.8), PII-Codex produces the candidate spans and types and the model filters and formats them, so +Tool measures a model$\times$library interaction.

**Skills-augmented (+Skills).** Same LangGraph (LangChain, 2024) agent with two additional tools: `list_skills` (returns available skill names) and `view_skill("pii-detection")` (returns the PII detection Skill document, `SKILL.md`). Models that read the Skill document before calling `analyze_pii` receive a structured prompt encoding PII-Codex tool syntax, output format, and type coverage. The same Skill document was used for both pilot and main studies. The document is deliberately minimal ($\sim$250 tokens) and points to the existing tool rather than adding new procedural capability; the +Skills condition therefore tests whether a lightweight tool-pointer skill adds value over tool access alone, not whether richer skill documents could.

**Baselines.** To separate the model's contribution from that of the underlying library, and to situate the four conditions against simpler interventions, three baselines are reported, scored under the identical alignment. *Detector-only* runs PII-Codex directly on the sample text (no model) and scores its raw output; this isolates the library. *Few-shot* (Brown et al., 2020) prepends in-context PII-annotated examples to the zero-shot prompt (single-turn, no tool). *Chain-of-thought* (Kojima et al., 2022) adds an explicit step-by-step reasoning instruction to the zero-shot prompt (single-turn, no tool). The few-shot and chain-of-thought baselines test whether the simplest parametric augmentations help where tool and skill scaffolding do not; both prompts are reproduced in Appendix I.

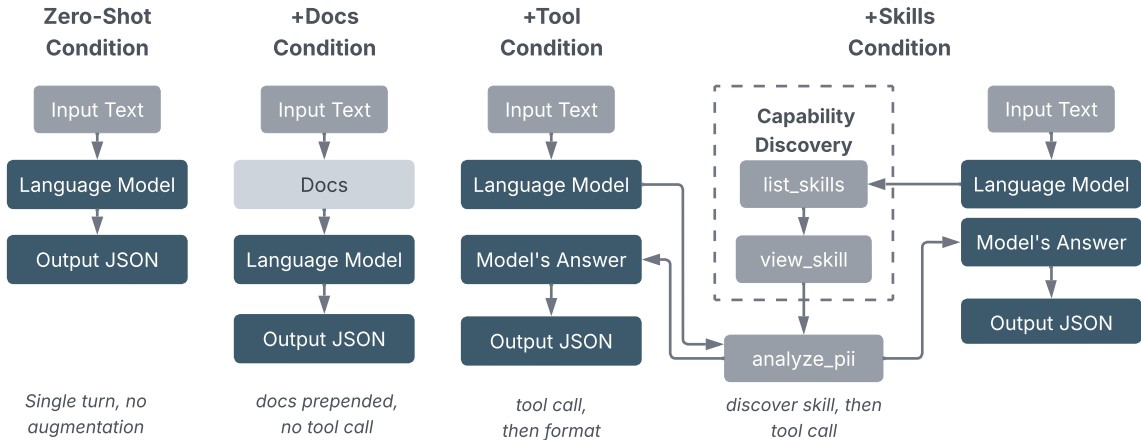

Figure 1: The four experimental conditions. Zero-shot is single-turn with no augmentation. +Docs prepends the PII-Codex reference documentation with no tool execution. +Tool grants access to `analyze_pii`, whose output the model filters and formats. +Skills adds a discovery loop (`list_skills`, `view_skill`) before the same tool call. All four conditions return a JSON array of PII predictions.

## 3.4 Execution infrastructure

Conditions were run sequentially per model (one condition at a time). Within multi-turn conditions (+Tool, +Skills), samples were processed in parallel via a ThreadPoolExecutor (1 worker in this study), with a shared lock serializing `generate` calls. Context was bounded per turn (initial user prompt + current system message only) to prevent context growth from degrading performance in the skills discovery loop. A per-sample timeout prevented runaway inference without discarding partial results.

## 3.5 Scoring and label alignment

Predictions were scored against ground truth using precision, recall, and F1. Two complementary matching criteria are reported as co-primary metrics. *Type-level* scoring counts a prediction as a true positive when its canonical type equals that of an as-yet-unmatched ground-truth entity; *span-level* scoring additionally requires the predicted and ground-truth character spans to overlap with intersection-over-union (IoU) $\geq$ 0.5. The two answer different questions (type-level asks whether the correct categories of PII present were identified, span-level whether they were also localized) and, as Section 4.1 shows, they do not always agree.

**Symmetric label alignment.** A single canonical normalization map $\phi\colon \mathcal{Y} \to \mathcal{Y}_{\text{canon}}$ is applied identically to predictions and ground truth in every condition and baseline. Built on a 136-entry base table (`pii_label_to_piicodex.json`), $\phi$ maps model-native label names (e.g., `person_name`, `ssn`, `date_of_birth`) to PII-Codex canonical types and bridges equivalent canonical types emitted by the detector versus recorded in the benchmark: most importantly `DATE` $\equiv$ `DATE_TIME` and `ADDRESS` $\equiv$ `LOCATION`, together with health-identifier variants. Aligned predictions are $\tilde{G}_i = \{(s, \phi(y)) \mid (s, y) \in \hat{G}_i\}$ and aligned ground truth is $\{(s, \phi(y)) \mid (s, y) \in G_i\}$; matching is performed in this shared canonical space.

Applying $\phi$ to only one side is unfair by construction. The tool and skills conditions invoke PII-Codex directly and emit canonical types that score without remapping. Remapping only zero-shot model-native labels, while leaving the tool's `DATE` output unbridged to the benchmark's `DATE_TIME`, therefore depresses tool and skills recall on temporal types for schema reasons and inflates zero-shot scores. All results below use the symmetric map $\phi$ and are *post-alignment*; pre-alignment scores are not comparable across conditions and are not reported.

**Span annotation coverage.** Span-level scoring presumes character offsets in the ground truth, and only one of the three benchmark sources provides them. Of the 2,000 samples, 1,132 (56.6%, all from `ai4privacy`) carry character spans; the remaining 868 (43.4%, from `gretel_pii_masking` and `nvidia_nemotron_pii`) are annotated by type alone. Where gold offsets are absent, the IoU constraint cannot be evaluated, so span-level scoring reduces to type-level scoring on those samples. The aggregate span-level figures reported in Section 4.1 are therefore a mixture of span-scored and type-scored items, which understates the true span-level separation between conditions. Section 4.2 reports the undiluted result on the span-bearing subset.

**Strict entity-level matching.** Alongside the two co-primary metrics, a standard *strict* entity-level criterion is reported, requiring an exact boundary match ($\hat{s} = s$ for both endpoints) in addition to type equality. Strict scoring separates approximate localization from exact offset generation, a distinction that Section 4.2 shows to be decisive.

### 3.6   Statistical analysis

The analysis uses a within-subjects, repeated-measures design, in which each of the 2,000 benchmark samples appears in all four conditions, and condition effects are estimated from paired within-sample differences $\Delta_i(c) = \mathrm{F1}(\tilde{G}_{i,c}, G_i) - \mathrm{F1}(\tilde{G}_{i,\mathrm{ZS}}, G_i)$. Reported are paired t-tests (condition F1 vs. zero-shot F1 at the sample level) and Cohen's $d$ for key comparisons (+Docs vs. ZS, +Tool vs. ZS, +Skills vs. ZS, and +Skills vs. +Tool). Bootstrap 95% confidence intervals are reported for mean F1 per (model, condition) cell (Appendix C, Table 16). Sample size was confirmed sufficient for the target margin of $\pm 0.05$ in all 16 cells (maximum required $n = 195$; actual $n = 2{,}000$). Decoding is greedy (argmax; `do_sample = False`), so model outputs are deterministic given a prompt. The reported effects are not subject to run-to-run generation variance, and the random seed governs only the fixed stratified subsample, not the generations themselves.

## 4   Results

### 4.1   Aggregate performance

Table 3 presents mean F1 (macro-averaged across types, and net of the label-schema artifact via the symmetric alignment) by model and condition under *type-level* scoring. Zero-shot leads both tool-augmented conditions for all four models; +Tool and +Skills underperform zero-shot by 6–17 percentage points. Documentation is largely neutral (Gemma, Mistral, Qwen: $\Delta \approx 0$) but substantially degrades Llama 3.1 8B ($\Delta = -0.15$). Figure 2 visualizes F1 by configuration under both metrics.

**The ranking depends on evaluation granularity.** Type-level and span-level scoring disagree on whether augmentation helps (Table 4; Figure 2). Under *span-level* matching (IoU $\geq 0.5$), the ordering inverts. Averaged across the four models, +Tool reaches F1 = 0.46 against zero-shot's 0.33. This inversion does not reflect better localization by the detector. As the decomposition below shows, it reflects the fact that the detector returns character offsets programmatically while the model must generate them. The aggregate figures are additionally diluted, since only 56.6% of samples carry gold offsets (Section 3.5). The "zero-shot wins" result, therefore, holds only at the type level; whether agentic augmentation helps is a function of the evaluation criterion, not of the scaffolding alone. Both metrics are reported as co-primary throughout.

**A model–library interaction.** Run standalone on the same texts, PII-Codex scores type-level F1 0.28 and span-level 0.23 (Table 4, "Detector"), below every model condition, including +Tool. The +Tool result is therefore a model×library interaction, not the library in isolation. The model nearly doubles the raw library's type-level F1 by filtering its over-predictions. The ordering is consistent. The model alone (zero-shot, 0.63) beats the model with the library (+Tool, 0.53), which beats the library alone (0.28). Adding the raw library output lowers type-level F1 even as the model improves on it. The library's apparent span-level lead is not a detection advantage. It recovers types at essentially the same rate as the model, and differs only in that its offsets are returned programmatically rather than generated. This ordering is unchanged without quantization and persists one model generation up: for Qwen2.5-14B, zero-shot again leads +Tool at the type level (0.61 versus 0.52; Section 5.6, Appendix F).

**Few-shot and chain-of-thought baselines.** Averaged across the four models, few-shot prompting matches zero-shot at the type level (0.628 vs. 0.631) and chain-of-thought is slightly worse (0.606), though per-model effects are small and mixed; the span-level picture is the same (few-shot 0.319, CoT 0.298, vs. zero-shot 0.325; Appendix C, Table 15). Per model, the effect is small and mixed (the few-shot delta ranges from $-0.065$ for Llama to $+0.032$ for Mistral). In-context *parametric* augmentation is roughly neutral ($|\Delta| \leq 0.03$ on average), whereas external tool and skill augmentation costs 0.10–0.17 F1. The large type-level cost is therefore specific to integrating the rule-based library, not to augmentation in general, and the zero-shot lead is not an artifact of comparing against a weak augmentation. Figure 3 places every configuration on a single type-level axis.

Table 3: Main study mean F1 by model and condition (N = 2,000, post-label alignment). All deltas are relative to zero-shot.

| Model | ZS | +Docs | $\Delta_{\text{Docs}}$ | +Tool | $\Delta_{\text{Tool}}$ | +Skills | $\Delta_{\text{Skills}}$ |
|---|---|---|---|---|---|---|---|
| Gemma 2 9B | 0.62 | 0.64 | $+0.01$ | 0.53 | $-0.10$ | 0.53 | $-0.10$ |
| Llama 3.1 8B | 0.68 | 0.53 | $-0.15$ | 0.57 | $-0.11$ | 0.50 | $-0.17$ |
| Mistral 7B | 0.58 | 0.59 | $+0.01$ | 0.53 | $-0.06$ | 0.53 | $-0.06$ |
| Qwen 2.5 7B | 0.64 | 0.65 | $+0.01$ | 0.49 | $-0.15$ | 0.48 | $-0.16$ |

Table 4: Mean F1 averaged across the four models, under the two co-primary metrics (main study, $N = 2,000$, symmetric alignment), with PII-Codex run standalone (no model) as a library-only reference. Type-level and span-level scoring disagree on the ranking of zero-shot versus the augmented conditions, and the standalone detector trails every model condition under both metrics.

| Metric | ZS | +Docs | +Tool | +Skills | Detector |
|---|---|---|---|---|---|
| Type-level | **0.63** | 0.60 | 0.53 | 0.51 | 0.28 |
| Span-level (IoU $\geq 0.5$) | 0.33 | 0.30 | **0.46** | 0.45 | 0.23 |

## 4.2 What span-level scoring measures: offset provenance

The span-level inversion in Section 4.1 might suggest that detector-backed pipelines localize PII better. Span-level scoring in fact conflates three distinct abilities. Decomposing it shows the inversion is driven entirely by the last of them, offset provenance.

For each condition, recall over span-bearing gold entities is decomposed into four nested stages: the prediction names the correct *type*; its span overlaps the gold entity at all (IoU $> 0$); it localizes the entity (IoU $\geq 0.5$); and it reproduces the exact boundary. The funnel is reported as entity-instance recall, pooled over gold entities, rather than the per-sample macro F1 used elsewhere in this paper, since the question here is what fraction of the PII present is recovered at each stage. Table 5 and Figure 4 report the result.

Type recovery is at parity across the conditions. Zero-shot recovers 27.2% of span-bearing gold entities by type, against 27.0% for +Tool, so tool augmentation does not help these models find more PII. The divergence is entirely positional. Conditioned on having named an entity correctly, model-generated offsets overlap it only 23.7% of the time (zero-shot) against 73.1% when the detector supplies them, and conditioned on overlapping, they reproduce the exact boundary 0.5% of the time against 82.8%. Exact-boundary matching under model-only prompting is effectively zero, at 0.0% to 0.1%.

The prompt asks the model to return each span's character `start` and `end` (Appendix I), and producing them requires counting characters in the input, which these models do poorly. The detector, by contrast, computes offsets programmatically during execution, and the model in the +Tool and +Skills conditions transcribes them. At the span level the two conditions are therefore evaluated on different tasks. The zero-shot model must generate the offsets; the tool-augmented pipeline inherits them.

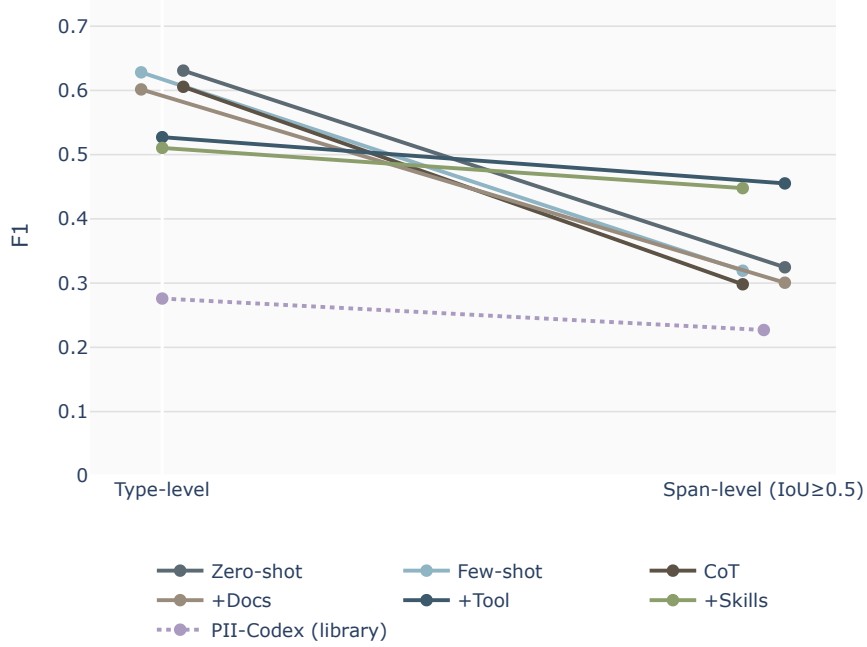

Figure 2: Type-level versus span-level F1 by configuration (four-model average, main study, $n = 2,000$, symmetric alignment). Each line connects a configuration's type-level F1 to its span-level F1 (IoU $\geq 0.5$); the crossing lines are the ranking inversion. The rise on span is an offset-provenance effect, not better detection (Section 4.2).

Table 5: Nested recall over the 28,448 span-bearing gold entities per condition (four-model total, symmetric alignment). The first four columns give recall of all gold entities at each successive stage. The last two are conditional, giving the share of correctly typed entities whose span also overlapped, and the share of overlapping spans that reproduced the exact boundary.

| Condition | Recall of gold entities | | | | Retained from previous stage | |
| | Type | Overlap | IoU$\geq$0.5 | Exact | of typed | of overlapped |
| --- | --- | --- | --- | --- | --- | --- |
| Zero-shot | 27.2% | 6.5% | 2.3% | 0.0% | 23.7% | 0.5% |
| +Docs | 25.7% | 5.7% | 2.0% | 0.1% | 22.3% | 1.0% |
| +Tool | 27.0% | 19.7% | 17.9% | 16.4% | 73.1% | 82.8% |
| +Skills | 26.0% | 19.2% | 17.6% | 16.1% | 74.0% | 83.9% |

Restricting to the span-bearing subset, where span scoring is not diluted by type-only items (Section 3.5), makes the size of the effect plain (Table 6). Zero-shot span-level F1 falls to 0.040 and strict F1 to 0.001, while type-level F1 remains competitive at 0.581. The aggregate span-level figures reported earlier (0.325 versus 0.455) substantially understate this separation, because 43.4% of the samples contribute type-level scores to the span-level average. The collapse is uniform across models, not an artifact of one weak model. Zero-shot exact-boundary F1 ranges only from 0.0002 to 0.0012 across the four (per-model values in Appendix E, Table 18).

The span-level "tool advantage" is thus a statement about offset provenance, not detection capability. It is the second asymmetry between model output and tool output surfaced here, alongside the label-ontology asym-

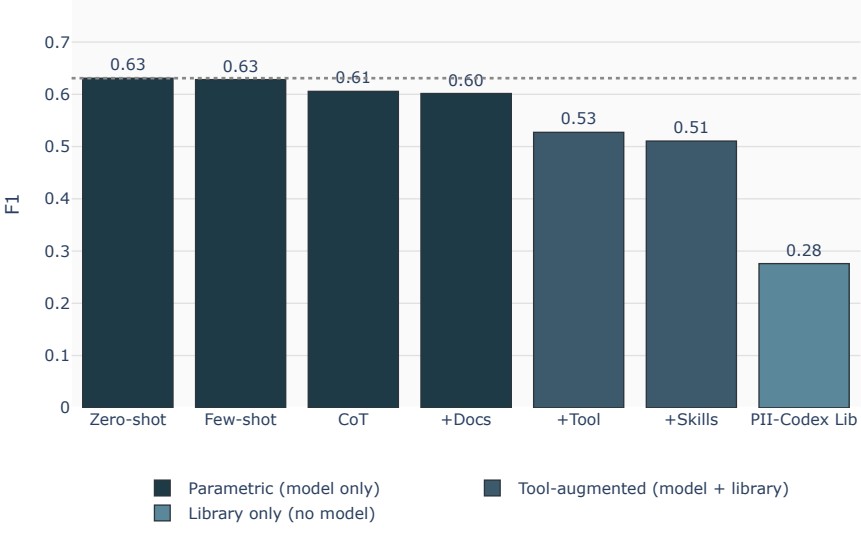

Figure 3: All configurations at a glance: type-level F1 averaged across the four models (main study, $n = 2{,}000$, symmetric alignment), grouped by family. The dashed line marks the zero-shot reference.

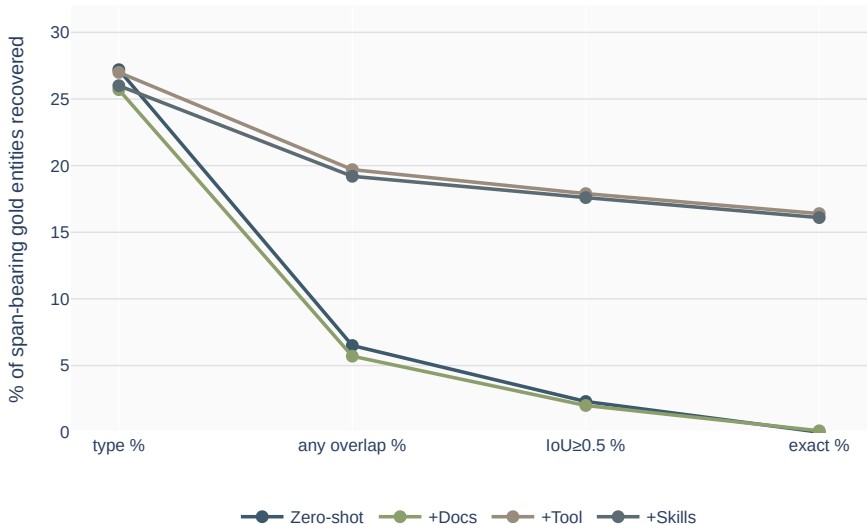

Figure 4: The detection funnel by condition. All four conditions recover PII types at parity; the parametric conditions then collapse at every stage that requires character offsets, while the detector-backed conditions retain most of their type recall through to exact-boundary matching.

metry of Section 3.5. This behavior is not unique to PII-Codex, as exposing structured offsets is a standard feature of the detector contract. Amazon Comprehend returns a zero-based `BeginOffset` and `EndOffset` per entity (Amazon Web Services, 2025), and Google Cloud Sensitive Data Protection a `codepointRange` per finding (Google Cloud, 2025). Any model asked for the same offsets must derive them by counting characters. The asymmetry therefore belongs to the interface between detector and model. One returns offsets as data; the other must generate them.

Table 6: Mean F1 by condition under three matching criteria (four-model average, symmetric alignment), on all samples and on the span-bearing subset ($n = 1{,}132$) where span scoring is meaningful. Type-level performance is preserved for the parametric conditions; span-level and strict performance is not.

| **Condition** | **All samples** ($n = 2{,}000$) | | | **Span-bearing** ($n = 1{,}132$) | | |
| | Type | Span | Strict | Type | Span | Strict |
|---|---|---|---|---|---|---|
| Zero-shot | 0.631 | 0.325 | 0.303 | 0.581 | 0.040 | 0.001 |
| +Docs | 0.602 | 0.301 | 0.282 | 0.566 | 0.034 | 0.001 |
| +Tool | 0.527 | 0.455 | 0.435 | 0.520 | 0.393 | 0.356 |
| +Skills | 0.511 | 0.448 | 0.429 | 0.500 | 0.389 | 0.355 |

### 4.3 Sensitivity to the alignment map

The canonical map $\phi$ bridges labels that the benchmark and the detector name differently, so it is reasonable to ask whether the reported inversion is an artifact of those bridges. The two consequential ones, `ADDRESS` $\equiv$ `LOCATION` and `DATE` $\equiv$ `DATE_TIME`, were removed independently and jointly, and both co-primary metrics recomputed (Table 7, Figure 5). Removing the bridges lowers absolute F1 for every condition and narrows the span-level gap, from $-0.131$ to $-0.031$, but the type-level and span-level gaps never change sign. Zero-shot leads on type and trails on span under all four maps. The ranking inversion is therefore a property of the metrics, not of the alignment choice.

Table 7: Alignment-map sensitivity: zero-shot minus +Tool F1 under each metric, with the two consequential label bridges toggled (four-model average, $n = 2{,}000$). The inversion (positive type gap, negative span gap) holds under every map.

| **Label map** | **Type ZS** | **Type +Tool** | **Type gap** | **Span gap** | **Inverts?** |
|---|---|---|---|---|---|
| Full map | 0.631 | 0.527 | +0.104 | −0.131 | yes |
| No `ADDRESS`≡`LOCATION` | 0.611 | 0.484 | +0.127 | −0.102 | yes |
| No `DATE`≡`DATE_TIME` | 0.595 | 0.428 | +0.168 | −0.060 | yes |
| Neither bridge | 0.576 | 0.385 | +0.191 | −0.031 | yes |

### 4.4 Statistical significance and effect sizes

Table 8 reports paired t-test results and Cohen's $d$ for each model and condition against zero-shot, under type-level scoring. All +Tool and +Skills comparisons are statistically significant ($p < 0.0001$) with small-to-medium negative effect sizes ($d$ from $-0.16$ to $-0.49$). Documentation (+Docs vs. zero-shot) has no significant effect for Gemma, Mistral, or Qwen ($p = 0.07$, $0.18$, $0.09$; small positive $d$), and significantly degrades only Llama 3.1 8B ($p < 0.0001$, $d = -0.37$). Under an asymmetric alignment, these effect sizes are larger (for example, +Skills reaches $d = -0.67$ for Llama). The fair, symmetric alignment substantially reduces the apparent magnitude of degradation, while its direction and significance hold for every model.

The incremental effect of adding the Skill document on top of tool access (+Skills vs. +Tool) yields Cohen's $d$ of $-0.29$ for Llama, $-0.02$ for Mistral, $-0.02$ for Qwen, and $+0.04$ for Gemma. None reach conventional thresholds for a meaningful effect, confirming that the Skill document provides no consistent incremental benefit beyond tool access alone.

### 4.5 Per-PII-type recall

Aggregate F1 masks heterogeneous effects across PII types, and the per-type pattern is what separates a label-schema artifact from genuine detector behavior. Table 9 shows mean recall delta (+Skills vs. zero-shot) by type and model under the symmetric alignment; Figure 6 contrasts tool/skill recall by PII type before

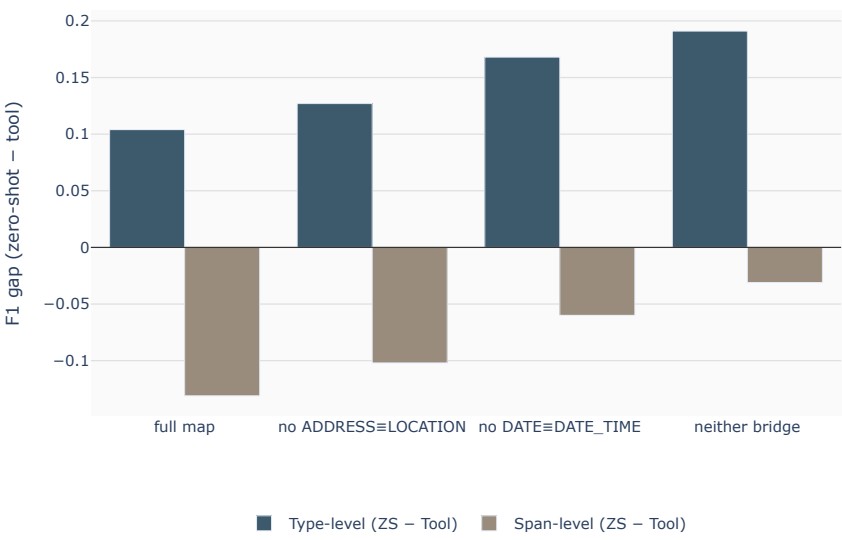

Figure 5: Type-level and span-level F1 gaps (zero-shot minus +Tool) under four alignment maps. The gaps shrink as bridges are removed but never cross zero, so the inversion does not depend on the bridging decisions.

Table 8: Paired t-test results and Cohen's $d$ (condition vs. zero-shot, N = 2,000 per model).

| Model | Condition | $t$ | $p$ | $d$ |
|---|---|---|---|---|
| Llama 3.1 8B | +Docs | $-16.50$ | $< 0.0001$ | $-0.37$ |
| | +Tool | $-14.27$ | $< 0.0001$ | $-0.32$ |
| | +Skills | $-21.84$ | $< 0.0001$ | $-0.49$ |
| Mistral 7B | +Docs | $+1.35$ | $0.18$ | $+0.03$ |
| | +Tool | $-7.08$ | $< 0.0001$ | $-0.16$ |
| | +Skills | $-7.14$ | $< 0.0001$ | $-0.16$ |
| Qwen 2.5 7B | +Docs | $+1.68$ | $0.092$ | $+0.04$ |
| | +Tool | $-18.05$ | $< 0.0001$ | $-0.40$ |
| | +Skills | $-18.66$ | $< 0.0001$ | $-0.42$ |
| Gemma 2 9B | +Docs | $+1.81$ | $0.071$ | $+0.04$ |
| | +Tool | $-11.97$ | $< 0.0001$ | $-0.27$ |
| | +Skills | $-11.87$ | $< 0.0001$ | $-0.27$ |

and after alignment; and Figure 7 summarizes the same recall surface as a configuration-by-type matrix. Full recall-by-type values for every condition and baseline across all four models are in Appendix D (Table 17).

**Temporal and Geographic Types Labeling.** Under an asymmetric alignment, `DATE_TIME` and `LOCATION` recall collapses nearly to zero under tool conditions. This is an artifact of how labels are matched, not of detection failure. The tool emits `DATE` and `ADDRESS` for spans the benchmark records as `DATE_TIME` and

Table 9: Recall delta (+Skills vs. zero-shot) by PII type and model (main study, symmetric alignment), ordered from most-improved to most-degraded. Positive values indicate higher recall under skills injection; negative values indicate degradation.

| PII Type | Gemma 2 9B | Llama 3.1 8B | Mistral 7B | Qwen 2.5 7B |
|---|---|---|---|---|
| US_DRIVERS_LICENSE_NUMBER | +0.23 | +0.32 | +0.22 | +0.21 |
| DATE_TIME | +0.32 | +0.03 | +0.28 | +0.02 |
| IP_ADDRESS | +0.17 | −0.02 | +0.14 | +0.05 |
| EMAIL_ADDRESS | +0.02 | 0.00 | +0.05 | +0.01 |
| LOCATION | +0.03 | −0.13 | +0.08 | −0.21 |
| US_PASSPORT_NUMBER | −0.08 | +0.03 | −0.03 | +0.10 |
| US_SOCIAL_SECURITY_NUMBER | −0.13 | −0.20 | −0.28 | −0.22 |
| PERSON | −0.20 | −0.25 | −0.06 | −0.20 |
| PHONE_NUMBER | −0.25 | −0.22 | −0.23 | −0.36 |
| HEALTH_INSURANCE_ID | −0.71 | −0.73 | −0.62 | −0.74 |

LOCATION, which an unbridged map leaves unmatched. Under the symmetric map (DATE ≡ DATE_TIME, ADDRESS ≡ LOCATION) these recover: DATE_TIME recall under +Skills is neutral-to-positive (e.g. +0.32 Gemma, +0.28 Mistral), and LOCATION no longer collapses.

**Medical Identifiers Coverage Gap.** HEALTH_INSURANCE_ID recall, by contrast, remains near zero under tool conditions even after symmetric alignment (−0.62 to −0.74 across models). This represents a coverage gap rather than a schema mismatch: PII-Codex does not map any detector to the HEALTH_INSURANCE_ID type, as Presidio lacked a general health-insurance recognizer for these identifiers at the time of writing. The health-identifier rules provided by Presidio are limited to specific formats (US Medicare MBI, UK NHS, Australian Medicare, US medical license) and do not match the benchmark's identifiers. Consequently, the tool misses detections that could be made from context. This situation highlights the difference between an evaluation artifact and a substantive failure mode.

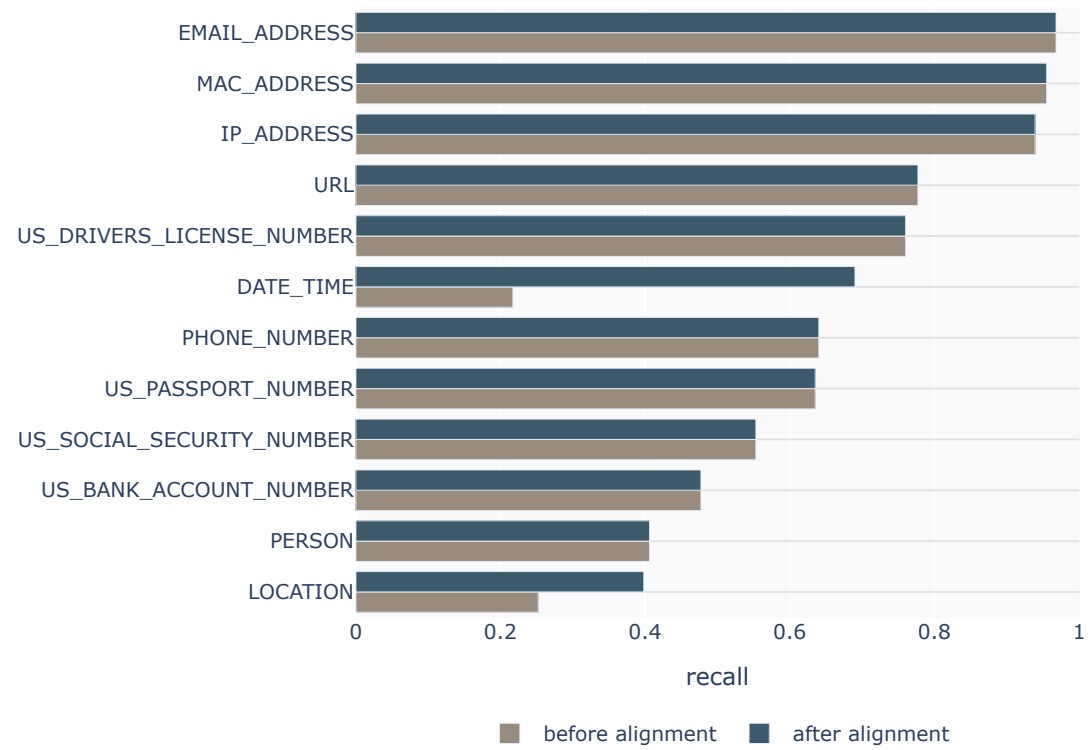

Figure 6: Tool/skill recall by PII type, before vs after symmetric alignment (main study, $n = 2{,}000$). `DATE_TIME` and `LOCATION` recover once the detector's `DATE/ADDRESS` outputs are bridged to the benchmark schema (the artifact); other types are unchanged. `HEALTH_INSURANCE_ID` remains near zero (a genuine detector limit).

**Recall Gains at a Precision Cost.** Types with unambiguous surface forms (`US_DRIVERS_LICENSE_NUMBER`, `IP_ADDRESS`) gain recall under tool use because the detector reliably flags them, though these gains are cancelled in aggregate by losses on context-dependent types, leaving overall type recovery at parity (Section 4.2). Its recall-first design also proposes these labels well beyond their true frequency (Section 4.6), so under exact-match F1 the recall gain is paid for in precision. Common semantic types (`PERSON`, `PHONE_NUMBER`, `US_SOCIAL_SECURITY_NUMBER`) lose recall under augmentation, as the model defers to tool output that misses spans it would otherwise have extracted from context.

### 4.6 Recall-oriented detection meets a precision-weighted metric

The recall gains on structured identifiers come with a precision cost under the exact-match F1 objective used here. Rule-based privacy detectors such as PII-Codex and the Presidio engine beneath it are deliberately tuned for high recall. In a redaction or compliance setting, over-flagging a candidate is far safer than missing real PII, so they propose candidates liberally by design. Aggregated over the tool and skills conditions, the detector therefore surfaces many identifier candidates: `US_BANK_ACCOUNT_NUMBER` 3,421 times against 44 true entities, `URL` 4,501 against 92, and `US_DRIVERS_LICENSE_NUMBER` 12,150 against 383 (Figure 8). This recall-first behavior is appropriate for the library's intended use, but a precision-weighted F1 penalizes it, and a *non-fine-tuned* model does not down-weight the extra candidates before emitting a final answer. The resulting precision cost is an integration effect, not a detection failure. Raw, recall-oriented detector output is passed to a model not trained to calibrate it, and this is the principal driver of the F1 decline that

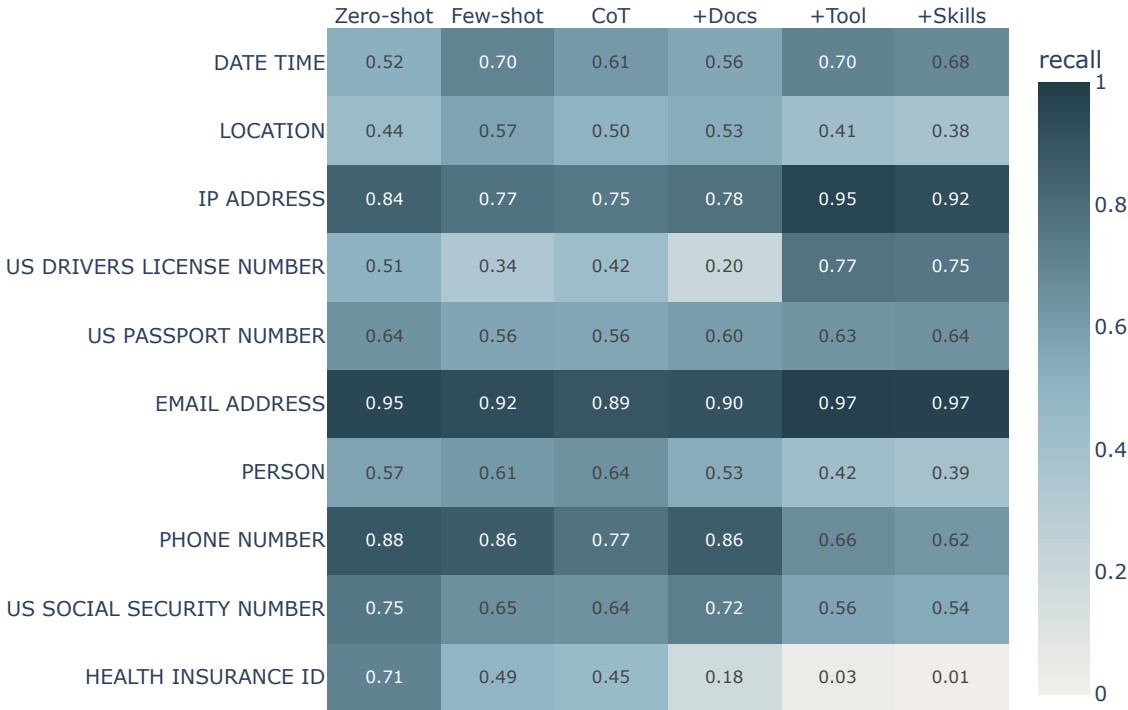

Figure 7: Recall by PII type and configuration (four-model average, main study, symmetric alignment). Structured identifiers brighten under +Tool/+Skills; `DATE_TIME` and `LOCATION` recover after alignment; `HEALTH_INSURANCE_ID` stays near zero under tool conditions (a genuine detector gap).

remains after symmetric alignment. It is not a deficiency of the library, whose span-level scores reflect offset provenance and not superior detection (Section 4.2).

## 4.7 Skill-viewed subgroup analysis

For the +Skills condition, Mistral read the Skill document in 948 of 2,000 runs (47.4%) and Qwen in 341 of 2,000 runs (17.1%). Llama and Gemma read the document in 100% of runs (no non-viewed comparison available). This spread reflects how consistently each model engaged the multi-step discovery protocol before answering: Mistral and Qwen often called `analyze_pii` without first reading the Skill document, consistent with the known difficulty non-fine-tuned 7–9B models have with multi-step tool-discovery loops (Patil et al., 2024; Qin et al., 2024). For Mistral (viewed $n = 948$, not-viewed $n = 1,052$) and Qwen (viewed $n = 341$, not-viewed $n = 1,659$) mean F1 was visually indistinguishable between the two groups in both cases (Appendix G, Table 21). This within-condition comparison controls for all factors except document reading, and provides no evidence that reading the Skill document improves PII detection performance, even when the model chooses to access it. Llama and Gemma read the document in every run, so the subgroup contrast is available only for Mistral and Qwen; it is therefore treated as suggestive rather than conclusive.

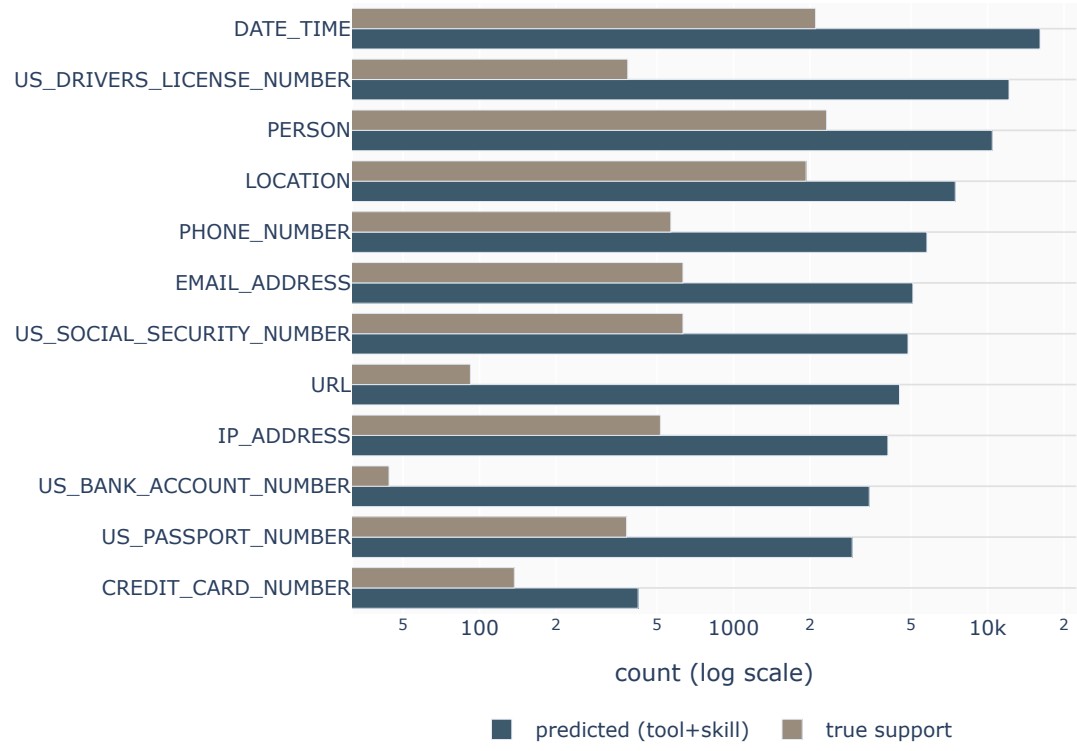

Figure 8: Predicted label volume versus true support in the tool/skills conditions (main study, log scale). The recall-tuned detector proposes identifier candidates well above their true frequency (`US_BANK_ACCOUNT_NUMBER` at 78×, `URL` at 49×, `US_DRIVERS_LICENSE_NUMBER` at 32×).

### 4.8 Execution reliability

Tool execution was reliable across both tool conditions. The 4,000 executions per model (2,000 samples × 2 conditions: +Tool and +Skills) completed successfully for Gemma, Llama, and Qwen; Mistral completed 3,998/4,000 (99.95%). Error rate was 0% across all 16 model-condition cells in both pilot and main study runs. The model emitted a tool call in essentially every run (two omissions in 16,000 +Tool and +Skills runs), so every "tool" run reflects genuine tool use, and the F1 degradation under tool conditions reflects tool-output quality and over-prediction. Pilot F1 summaries are reported separately in Appendix B (Tables 11 and 12).

## 5 Discussion

### 5.1 The pre-alignment confound

The central result is partly methodological and partly behavioral, and disentangling the two is a key contribution of this study. The methodological part arises because tool conditions invoke PII-Codex directly and therefore output canonical types that score without remapping, while zero-shot outputs require a post-hoc alignment step. A study that reports results before label alignment would incorrectly conclude that tool use substantially improves PII detection. After alignment, the comparison is fair, and the direction of effect reverses (Figure 9, Asymmetry 1). Just as superficial metrics in model editing can mask underlying failures and yield systematically misleading conclusions about a technique's success (Baser et al., 2026), comparing

raw tool versus zero-shot scores without normalizing the label schema gives a false sense of tool superiority. This confound is not specific to this study's design; it is a general risk for any evaluation framework that mixes model-native and tool-canonicalized predictions without explicit normalization. Future benchmarking of agentic PII systems should therefore include a mandatory label-alignment pass and report pre- and post-alignment scores separately.

Label schema mismatch is only one asymmetry. A second is offset provenance (Section 4.2), which is not a labeling problem at all yet arises for the same reason. The two conditions emit outputs of a different form, and the evaluator compares them as though they did not. Figure 9 presents both, together with the parity result that anchors them. The conditions recover PII types at essentially the same rate, so neither asymmetry reflects a difference in what the system can detect. The generalizable point is developed in Section 5.5.

## 5.2  Why does tool use degrade aggregate F1?

The per-type analysis in Section 4.5 separates two contributions to the aggregate decline. One is an evaluation artifact, and one is substantive.

**The evaluation artifact removed by alignment.** In the tool and skills conditions, the detector emits the canonical type `DATE` for temporal spans the benchmark records as `DATE_TIME`, and `ADDRESS` for spans it records as `LOCATION`. Under an alignment map that does not bridge these equivalent types, the predictions go unmatched, and recall on two of the benchmark's highest-frequency types collapses to near zero, an artifact of the scoring schema, not of detection. The symmetric map (Section 3.5) bridges `DATE` ≡ `DATE_TIME` and `ADDRESS` ≡ `LOCATION`, and the collapse disappears. Under the augmented conditions, `DATE_TIME` recall becomes neutral-to-positive and `LOCATION` recovers (Table 9, Figure 6). The symmetric map cuts the mean zero-shot-over-tool gap by about 38% (from 0.169 to 0.104), so a substantial part of the aggregate type-level degradation is this scoring artifact and not a detection failure.

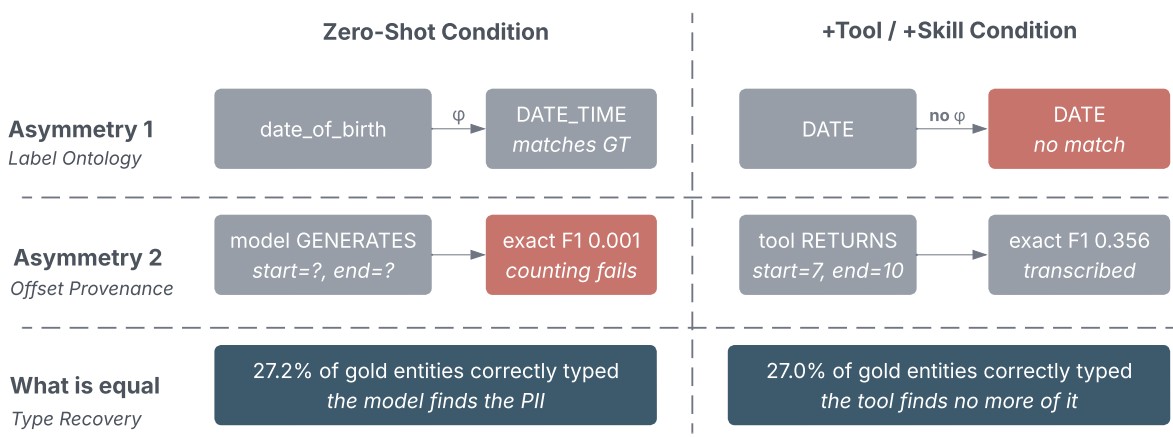

Figure 9: The two evaluation asymmetries. Type recovery is at parity (27.2% versus 27.0%), so the conditions detect equally well and differ only in the form of what they emit: label ontology, and offset provenance (exact-boundary F1 0.001 for the model, 0.356 for the tool). Each unreconciled asymmetry manufactures a capability difference that does not exist.

**The substantive effect that remains.** The residual gap after symmetric alignment is substantive, and it turns on context. The rule-based detector is tuned for high recall and proposes identifier labels well above their true frequency (Section 4.6). Under an exact-match, precision-weighted metric, a non-fine-tuned model that passes those candidates through uncalibrated pays for the extra volume in precision. And for contextually ambiguous types, most clearly `HEALTH_INSURANCE_ID`, the determination of whether a span is sensitive depends on surrounding text that a pattern-based detector does not see, so the models, reasoning from context, achieve higher recall than PII-Codex does in isolation, and routing through the tool can discard

those context-derived detections. This is consistent with PII-Codex's own framing, in which the sensitivity of a token is context- and use-dependent rather than a fixed property of its surface form (Rosado, 2023), and with subsequent work arguing that privacy-enhancing tools overlook the context-dependent nature of privacy risk when they assess identifiers in isolation rather than modeling the relationships among them (Zuo & Lee, 2026). As Section 4.1 shows, the standalone detector trails every model condition at the type level (0.28) and its span-level figure reflects offset provenance and not detection (Section 4.2). The comparison is a model–library interaction, and the question is how, not whether, to combine them.

This pattern mirrors knowledge-intensive settings where generated internal knowledge surpasses externally retrieved knowledge (Yu et al., 2023; Chen et al., 2023; Luo et al., 2025; Qian et al., 2025). External programmatic tools for PII lack the nuanced contextual understanding that neural representations possess, and the coherence of model-native output can outweigh the occasional benefit of rule-based canonicalization. Tools are warranted when models must integrate real-time or frequently updated information beyond their pretrained knowledge (Yu & Ji, 2024). PII detection is a static, pattern-recognition task on which these models are not improved by tool access, so the scaffolding adds interface cost without a corresponding gain. Models often lack reliable self-awareness of their knowledge boundaries (Yin et al., 2023), which may explain why they invoke tools even when tool output underperforms their own parametric knowledge.

**Type-level versus span-level.** The two co-primary metrics were intended to separate "finding the span" from "labeling it". The decomposition in Section 4.2 shows the span metric does not measure finding: type recovery is at parity, and the span-level advantage of the augmented conditions comes from who produces the offsets. Reporting both metrics is what makes that asymmetry legible; reporting the span metric alone would have credited the detector with a capability it does not have.

## 5.3 Documentation and the Llama exception

Three of four models show near-zero sensitivity to documentation injection ($|\Delta| \leq 0.01$, all non-significant). Llama 3.1 8B is a notable outlier, with a statistically significant $-0.15$ degradation under +Docs ($p < 0.0001$, $d = -0.37$). This is consistent with recent findings that a model's stronger baseline instruction-following ability can paradoxically degrade performance by making it overly sensitive to rigid or suboptimal formatting constraints in the prompt (Tam et al., 2024; Luo et al., 2025). Furthermore, injecting external documentation can create "merge conflicts" with a model's parametric knowledge (Qian et al., 2023). When given the PII-Codex documentation, Llama may attempt to strictly map its output to the documentation's label set rather than relying on its own more accurate internal representations.

## 5.4 Skill document reading does not explain performance

The skill-viewed subgroup analysis (Section 4.7) together with the null +Skills-vs-+Tool comparison (Table 8) indicates that the Skill document evaluated here adds no value beyond tool access. This conclusion is scoped to the specific ∼250-token document tested rather than to skill documents in general; a richer document with worked examples and type disambiguation might behave differently (see Limitations), and the subgroup contrast is available only for Mistral and Qwen. Within that scope, models that read the document do not outperform models that do not. The document's failure to help may reflect a context competition effect, in which the Skill document consumes context capacity that could otherwise be used for the model's own chain-of-thought or attention to the sample text; dense instructions and tool outputs in the context window trigger context competition, degrading a model's ability to extract and use relevant information (Liu et al., 2024; Luo et al., 2025; Xu et al., 2025; Shi et al., 2023). The +Skills condition also imposes a longer tool trajectory than +Tool, since `list_skills` and `view_skill` precede `analyze_pii`. Each additional step is another opportunity for a formatting or generation error in a 7–9B model, so part of any +Skills penalty may reflect this interaction overhead rather than the document's content. Turn count and elapsed time show at most weak correlation with per-sample F1 ($|r| \leq 0.16$; Appendix G, Table 22).

## 5.5 Implications for agentic NLP pipeline design

The results suggest a broader design principle. Agent scaffolding imposes structural constraints that can conflict with model representations. Varying prompt phrasings or rigid output constraints lead to divergent implementations and metrics, undermining the reliability of agent evaluation (Sun et al., 2025). These conclusions concern *non-fine-tuned* tool calling, and prior work shows that 7–9B base models require domain-specific fine-tuning for reliable API use (Patil et al., 2024), so this finding speaks to the default, out-of-the-box configuration practitioners are most likely to deploy, not to tool use after fine-tuning.

1. **Do not assume tool use improves structured extraction.** Consistent with tool-use literature (Luo et al., 2025; Qian et al., 2025; Xu et al., 2025), tool-augmented pipelines in the 7–9B parameter class were found to add latency and infrastructure complexity while reducing accuracy relative to zero-shot prompting with label normalization. Per-type recall gains on structured identifiers are cancelled in aggregate by losses on context-dependent types, leaving overall type recovery at parity with zero-shot (Section 4.2); tool access redistributes which PII is found rather than increasing how much is found.

2. **Label alignment is a first-class evaluation concern.** Any mixed-mode pipeline (some predictions from tool output, some from model output) must normalize labels before scoring. Omitting this step will produce systematically misleading comparisons.

3. **Offset provenance is a first-class evaluation concern.** A detector returns character offsets; a model must generate them, and at this scale effectively cannot (Section 4.2). A span-level metric therefore does not compare the two systems on the same task, and credits the tool-augmented pipeline for the harness. For category detection, meaning which PII types a document contains, type-level scoring is the appropriate criterion. For localization, meaning where the spans are, the offset-generation burden must first be equalized, for example by resolving model-predicted span text to offsets programmatically.

4. **Evaluate the asymmetries, not just the systems.** Both concerns above are one phenomenon. A mixed pipeline compares two producers that differ not only in competence but in the form of what they emit, namely label ontology and offset provenance. Each unreconciled asymmetry manufactures an apparent capability difference where none exists. Neither is peculiar to PII-Codex, since publishing structured offsets is standard in the detector contract (Amazon Web Services, 2025; Google Cloud, 2025). The methodological response is to enumerate and reconcile the asymmetries between two output formats before attributing any measured difference to the models.

5. **The effect is a pipeline property, not a capability or scale property.** One might attribute the degradation to model capability within the 7–9B class; the control experiments do not support that reading. The 14B model degrades by essentially the same margin (Section 5.6), so the gap is comparable one model generation up, and the rank ordering of degradation among the four 7–9B models is weak enough that it is not interpreted mechanistically. The effect is instead located in the pipeline (label schema, evaluation granularity, and the model–library interface), which is invariant to the capability differences observable here. Consistent with evidence that rigidly constraining output format can harm performance (Tam et al., 2024; Luo et al., 2025), agent scaffolding can conflict with a model's learned representations. On the present evidence, this reflects the interface and the evaluation, not model size (Sections 4.2 and 5.6).

## 5.6 Robustness: scale and precision

Two checks probe whether the pattern is an artifact of quantization or limited to the smallest models, and a third addresses generation stochasticity.

**Scale.** Re-running the full ablation on Qwen2.5-14B (4-bit) over the same 2,000 samples leaves the pattern essentially unchanged from the 7–9B average. At the type level, zero-shot still leads +Tool by a comparable margin (+0.09 versus a 7–9B average of +0.10), and at the span level the gap attributable to offset provenance

persists at a comparable magnitude ($-0.14$ versus $-0.13$), so the asymmetry is invariant to scale. The tool executed in 99.9% of these runs (3,997/4,000), so the gap persists despite near-perfect execution, and a larger model that invokes the tool more reliably does not make augmentation begin to help (Figure 10; full condition-by-condition values in Appendix F, Table 19). This scale check is a single model one generation above the studied class; broader sweeps (e.g., 32B and above) and other model families remain future work.

**Precision.** Re-running the four models unquantized (fp16/bf16) on a stratified 300-sample subset, compared per model against 4-bit on the identical samples, shows unquantized F1 within $0.006 \pm 0.006$ (type level) and $0.004 \pm 0.004$ (span level) of 4-bit, where each value is the mean per-sample fp16$-$4-bit F1 difference with its 95% CI; the largest single-model shift is $+0.028$ (Llama), an order of magnitude below the augmentation effects and changing no ordering. Quantization is therefore not responsible for the observed pattern (Figure 11; per-model values in Appendix F, Table 20).

**Decoding.** Decoding is greedy (Section 3.6), so model outputs are deterministic given a prompt; the reported effects are not subject to run-to-run generation variance, and a seed sweep is not applicable.

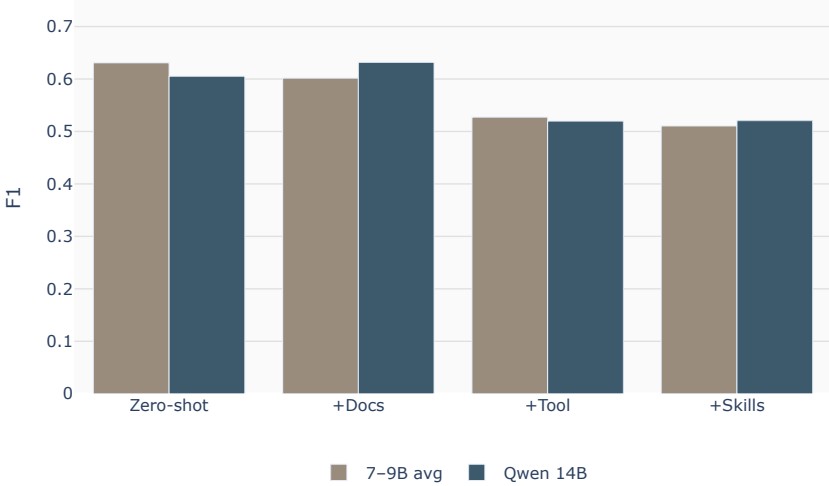

Figure 10: Type-level F1 by configuration for the 7–9B average versus Qwen2.5-14B (main study, symmetric alignment). The augmentation pattern is unchanged at 14B.

## 5.7 Limitations

This study is scoped to English-language text and US-locale PII types, so the findings may not transfer to other languages or international PII schemas. The benchmark covers 21 PII types from three sources, with limited coverage of rare types such as `ABA_ROUTING_NUMBER` and `LICENSE_PLATE_NUMBER`. The robustness checks are deliberately narrow. The primary runs use 4-bit quantization in the 7–9B class. The pattern holds without quantization (four models) and for a single 14B model (Qwen2.5-14B), but broader scale sweeps (32B and above) and additional model families remain future work, as do fine-tuned models. Two design choices bound how the tool result should be read. First, the Skill document is a single, deliberately minimal ($\sim$250-token) pointer to the existing tool, compressed to limit context overload in small models (Luo et al., 2025; Xu et al., 2025). A richer document with worked examples and type disambiguation may behave differently. Second, the comparison is *non-fine-tuned*, out-of-the-box tool calling, and foundational work shows that 7–9B base models generally fail to execute rigid API calls reliably without targeted fine-tuning (Patil et al., 2024; Qin et al., 2024), so the degradation reflects the default configuration practitioners are most likely to deploy rather than an indictment of tool use after fine-tuning. The low skill-document engagement rates for Qwen (17.1%) and Mistral (47.4%) are consistent with this. The agent also uses a lenient tool-call detection strategy. A stricter enforcement of tool use before accepting answers (available but not used in the default

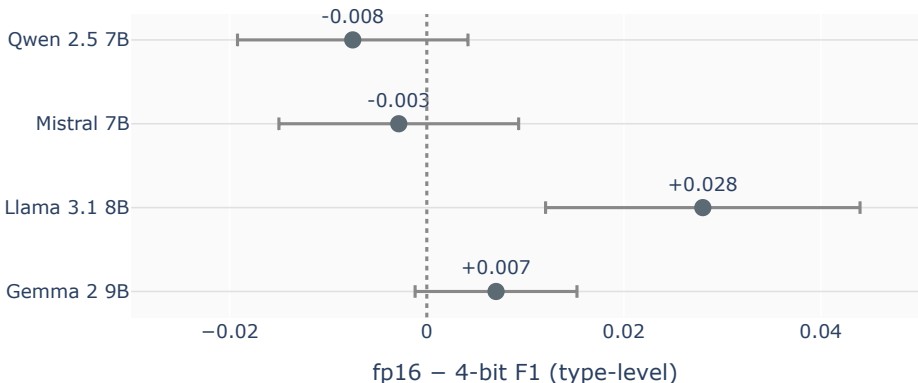

Figure 11: Per-model type-level F1 difference (unquantized minus 4-bit) on the shared 300-sample subset (mean ± 95% CI; dotted line marks zero). All four differences lie within 0.03 F1 of zero.

runner) may produce different results. The result is therefore best read as specific to this model class, skill construction, and task configuration.

Two limitations concern the offset-provenance analysis (Section 4.2). First, it establishes that model-generated offsets are almost never correct, but cannot separate a prediction whose text is right and whose offsets are wrong from one that is simply wrong, since predicted span text is redacted at write time to avoid persisting PII, leaving only types and offsets in the released records. Quantifying that split needs a re-run storing a privacy-preserving flag (whether predicted text matches the source at the predicted offsets) in place of the text, and is left to future work. Second, while the offset asymmetry is structural, and the detector contracts cited above indicate it generalizes, it is demonstrated here with one detector family and one non-fine-tuned integration, in which the model is instructed to rely on tool output. An integration permitting the model to accept, reject, or supplement detector predictions is a valuable future work item and is not evaluated here.

## 6 Conclusion

A controlled ablation evaluated documentation injection, tool access, and Agent Skills documents for PII detection across four open-weight SLMs, scored under a symmetric label alignment and two co-primary metrics. Whether augmentation helps depends on how detection is scored. At the entity-type level, zero-shot prompting outperforms both tool-augmented conditions for every model in the 7–9B class, with tool and skills reducing mean F1 by 6–17 points; at the span level the ranking reverses. Decomposing that reversal shows it is not a detection effect. The models identify PII types at parity with the detector-backed conditions, and diverge only once character offsets are required, where the detector returns them programmatically and the model, which must generate them, reaches an exact-boundary F1 of 0.001. The span metric rewards the harness, not detection. Averaged across models, few-shot and chain-of-thought prompting do not exceed zero-shot, and the skill adds nothing over tool access, so the type-level zero-shot lead is not an artifact of a weak baseline. The pattern persists for a 14B model and without quantization, which locates the effect in the evaluation pipeline and the non-fine-tuned model–library interface, and the pattern is not limited to the smallest models in the study. This is consistent with evidence that reliable tool use in this class requires fine-tuning (Schick et al., 2023; Patil et al., 2024).

Two artifacts drive the result, in label ontology and in offset provenance, and both are asymmetries between what a model emits and what a tool emits. The two producers differ not merely in competence but in the form of their output. Each unreconciled asymmetry manufactures an apparent capability difference where none exists, and both are structural properties of any model-plus-detector pipeline. The generalizable claim

is that such asymmetries must be enumerated and reconciled before any measured difference is attributed to the models. Future work should test additional detectors, integration strategies that permit the model to override tool output, richer skill documents, and PII schemas beyond the US-English scope studied here.

**Broader Impact Statement**

This work evaluates the effect of agent augmentation on PII detection accuracy in small language models, and its practical implication is cautionary in two directions. Practitioners deploying agentic scaffolding for sensitive data tasks should validate augmentation empirically rather than assume gains. Equally, much of the degradation observed here is an evaluation and label-alignment artifact, not evidence that tools are inherently worse at finding PII, so reporting "tool use degrades PII detection" without that context could lead deployers to abandon scaffolding for the wrong reason.

The converse error is the more consequential one for privacy. A detector supplies exact character offsets and a language model cannot, so span-level benchmarks systematically inflate the scores of tool-augmented pipelines. A practitioner selecting a redaction system on span scores may conclude that augmentation improved detection when it only supplied the character offsets the model could not generate. For redaction, the offsets are what get masked, so a pipeline that supplies them programmatically compensates for a limitation the model cannot overcome, and the type-level parity does not argue against it. For auditing or disclosure review, where the question is which categories of PII a document contains, that same span-level lead is illusory, and the simpler model-only pipeline performs comparably. Matching the metric to the deployment question is therefore a privacy-relevant decision, not merely a methodological one.

The benchmark, scored results, and analysis notebooks are publicly released to support reproducibility and to enable the community to verify, extend, or challenge these findings. All datasets used are synthetic or publicly available; no real personal data was used or exposed in this research.

## Data and Code Availability

Benchmark ground truth is available at [redacted]. Scored predictions and per-sample scores for all conditions, baselines, and robustness controls are available at [redacted]. The analysis notebooks (under `notebooks/`) and the full project repository are available at [redacted].

## Acknowledgments

The author used Claude Opus 4.8 (Anthropic) to assist with notebook restructuring, code refinement, and LaTeX manuscript preparation. All experimental design, data collection, analysis, and scientific conclusions are the author's own.

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

# A Benchmark Details

## A.1 Full PII Type Distribution

Table 10 presents the complete ground-truth entity count for all PII types present in main study benchmark. Counts are computed from the benchmark's `pii_codex_ground_truth` field.

Table 10: Full ground-truth PII entity counts by type (main study, N = 2,000). Each count is the number of entity spans of that type in the benchmark; total matches Table 1.

| PII Type | Entity count |
|---|---|
| PERSON | 2,324 |
| DATE_TIME | 1,572 |
| LOCATION | 1,068 |
| ADDRESS | 862 |
| EMAIL_ADDRESS | 632 |
| US_SOCIAL_SECURITY_NUMBER | 632 |
| PHONE_NUMBER | 566 |
| DATE | 531 |
| IP_ADDRESS | 515 |
| US_DRIVERS_LICENSE_NUMBER | 383 |
| GENDER | 365 |
| HEALTH_INSURANCE_ID | 343 |
| ZIPCODE | 316 |
| US_PASSPORT_NUMBER | 379 |
| CREDIT_CARD_NUMBER | 137 |
| LICENSE_PLATE_NUMBER | 175 |
| PASSWORD | 127 |
| OCCUPATION | 102 |
| URL | 92 |
| ABA_ROUTING_NUMBER | 62 |
| US_BANK_ACCOUNT_NUMBER | 44 |
| NRP | 28 |
| MEDICAL_LICENSE | 30 |
| RACE | 18 |
| AGE | 18 |
| SWIFT_CODE | 17 |
| MAC_ADDRESS | 14 |
| US_INDIVIDUAL_TAXPAYER_IDENTIFICATION | 11 |
| SEXUAL_PREFERENCE | 7 |
| **Total (entity span count)** | **11,370** |

## A.2 Labels Excluded from Scoring

The following 27 prediction labels appeared in model output but could not be unambiguously mapped to a PII-Codex canonical type and were excluded from scoring. They are listed here for transparency. `COUNTRY_OF_ORIGIN` was mapped to `LOCATION` and is not in this list.

```
ATTRIBUTE_NAME, ATTRIBUTE_VALUE, CLIENT_ID, COMMENTS, ID, IDENTIFIABLE,
IDENTIFICATION_NUMBER, IDENTIFIER, ID_NUMBER, INDIVIDUAL_ID,
MEDICALCONDITION, NUMBER, PARENT_ID, PARTY, PERSONAL_IDENTIFIER,
PERSON_ID, PII, PII_TYPE, PSYCHOLOGICAL_HISTORY, ROLE, SIBLING_ID,
SUBSTANCEUSEDISORDERHISTORY, TEXT, TRAUMAHISTORY, US_ID, US_ID_NUMBER,
VEHICLE_ID
```

Labels such as `ATTRIBUTE_NAME`, `COMMENTS`, `ROLE`, and `TEXT` do not appear in any of the three source datasets and are model-output artifacts rather than true PII categories. `VEHICLE_ID` was considered for mapping to `LICENSE_PLATE_NUMBER` but was held for manual review and not included in the canonical mapping.

# B Pilot Study Results

The pilot (n = 200) was run on the same hardware and under the same conditions as the main study, using a stratified subsample of the full benchmark. It served primarily to validate the pipeline and inform the main study configuration (parallelism, context bounds, timeouts). Pilot results are presented here as a citable record; formal conclusions are drawn from the main study only.

## B.1 Pilot Summary Table

Table 11: Pilot study summary (n = 200, post-label alignment). f1_med = median F1; elapsed_$\mu$ = mean seconds per sample; turns_$\mu$ = mean conversation turns.

| Model | Condition | f1_$\mu$ | prec_$\mu$ | rec_$\mu$ | f1_med | skill_viewed | elapsed_$\mu$ | turns_$\mu$ |
|-------|-----------|------|--------|-------|--------|--------------|-----------|----------|
| Gemma 2 9B | with_docs | 0.557 | 0.710 | 0.493 | 0.571 | 0 | 9.75 | 1.00 |
| | with_skills | 0.391 | 0.454 | 0.388 | 0.400 | 200 | 11.81 | 4.00 |
| | with_tools | 0.392 | 0.455 | 0.387 | 0.400 | 0 | 8.22 | 2.00 |
| | zero_shot | 0.530 | 0.683 | 0.466 | 0.571 | 0 | 7.60 | 1.00 |
| Llama 3.1 8B | with_docs | 0.412 | 0.446 | 0.419 | 0.453 | 0 | 8.56 | 1.00 |
| | with_skills | 0.384 | 0.419 | 0.409 | 0.400 | 200 | 15.74 | 3.93 |
| | with_tools | 0.433 | 0.467 | 0.456 | 0.437 | 0 | 14.91 | 2.00 |
| | zero_shot | 0.598 | 0.620 | 0.613 | 0.667 | 0 | 3.35 | 1.00 |
| Mistral 7B | with_docs | 0.503 | 0.629 | 0.446 | 0.500 | 0 | 4.92 | 1.00 |
| | with_skills | 0.434 | 0.534 | 0.406 | 0.400 | 108 | 9.63 | 2.02 |
| | with_tools | 0.434 | 0.534 | 0.406 | 0.400 | 0 | 10.94 | 2.00 |
| | zero_shot | 0.497 | 0.628 | 0.438 | 0.571 | 0 | 3.04 | 1.00 |
| Qwen 2.5 7B | with_docs | 0.543 | 0.646 | 0.500 | 0.571 | 0 | 4.78 | 1.00 |
| | with_skills | 0.394 | 0.543 | 0.361 | 0.400 | 43 | 12.99 | 2.47 |
| | with_tools | 0.392 | 0.461 | 0.391 | 0.400 | 0 | 7.28 | 2.02 |
| | zero_shot | 0.556 | 0.634 | 0.528 | 0.608 | 0 | 3.01 | 1.00 |

### B.2 Pilot Degradation Tabular Heatmap

Table 12: Pilot mean F1 delta vs. zero-shot by model and condition.

| Model | $\Delta$+Docs | $\Delta$+Tool | $\Delta$+Skills |
|---|---|---|---|
| Gemma 2 9B | **+0.028** | −0.138 | −0.139 |
| Llama 3.1 8B | −0.186 | −0.165 | −0.214 |
| Mistral 7B | **+0.005** | −0.063 | −0.063 |
| Qwen 2.5 7B | −0.013 | −0.164 | −0.162 |

## C Full Statistical Tables

### C.1 All Pairwise Comparisons

Table 13 extends Table 8 in the main text to include all six pairwise comparisons per model. Cohen's $d$ and paired $t$ and $p$ are reported for all comparisons.

Table 13: Full pairwise delta summary (main study, N = 2,000 per model). % imp = percentage of samples where the condition outperforms the comparator.

| Model | Comparison | $t$ | $p$ | $d$ | % improved |
|---|---|---|---|---|---|
| Llama 3.1 8B | +Docs vs ZS | −16.50 | < 0.0001 | −0.369 | 24.4% |
| | +Tool vs ZS | −14.27 | < 0.0001 | −0.319 | 25.2% |
| | +Skills vs ZS | −21.84 | < 0.0001 | −0.488 | 22.0% |
| | +Tool vs +Docs | +4.66 | < 0.0001 | +0.104 | 41.8% |
| | +Skills vs +Docs | −2.56 | 0.011 | −0.057 | 35.8% |
| | +Skills vs +Tool | −12.90 | < 0.0001 | −0.288 | 11.4% |
| Mistral 7B | +Docs vs ZS | +1.35 | 0.179 | +0.030 | 33.0% |
| | +Tool vs ZS | −7.08 | < 0.0001 | −0.158 | 35.2% |
| | +Skills vs ZS | −7.14 | < 0.0001 | −0.160 | 35.2% |
| | +Tool vs +Docs | −8.21 | < 0.0001 | −0.184 | 33.5% |
| | +Skills vs +Docs | −8.26 | < 0.0001 | −0.185 | 33.1% |
| | +Skills vs +Tool | −0.75 | 0.453 | −0.017 | 1.6% |
| Qwen 2.5 7B | +Docs vs ZS | +1.68 | 0.092 | +0.038 | 30.8% |
| | +Tool vs ZS | −18.05 | < 0.0001 | −0.404 | 23.9% |
| | +Skills vs ZS | −18.66 | < 0.0001 | −0.417 | 23.6% |
| | +Tool vs +Docs | −21.41 | < 0.0001 | −0.479 | 22.8% |
| | +Skills vs +Docs | −22.87 | < 0.0001 | −0.511 | 21.9% |
| | +Skills vs +Tool | −0.68 | 0.496 | −0.015 | 26.1% |
| Gemma 2 9B | +Docs vs ZS | +1.81 | 0.071 | +0.040 | 29.9% |
| | +Tool vs ZS | −11.97 | < 0.0001 | −0.268 | 29.4% |
| | +Skills vs ZS | −11.87 | < 0.0001 | −0.266 | 29.4% |
| | +Tool vs +Docs | −14.81 | < 0.0001 | −0.331 | 26.9% |
| | +Skills vs +Docs | −14.72 | < 0.0001 | −0.329 | 27.1% |
| | +Skills vs +Tool | +1.79 | 0.074 | +0.040 | 1.2% |

### C.2   Parametric baselines (few-shot, chain-of-thought)

Table 14 reports the paired comparison of the few-shot and chain-of-thought baselines against zero-shot at the type level. Per model, the effect is mixed; averaged across the four models, neither parametric baseline exceeds zero-shot (Section 4.1).

Table 14: Few-shot and chain-of-thought baselines versus zero-shot (main study, $N = 2{,}000$ per model, type-level). % improved = percentage of samples where the baseline outperforms zero-shot.

| Model | Comparison | $t$ | $p$ | $d$ | % improved |
|---|---|---|---|---|---|
| Llama 3.1 8B | Few-shot vs ZS | $-8.91$ | $< 0.0001$ | $-0.199$ | 27.0% |
| | CoT vs ZS | $-13.29$ | $< 0.0001$ | $-0.297$ | 24.1% |
| Mistral 7B | Few-shot vs ZS | $+4.61$ | $< 0.0001$ | $+0.103$ | 41.2% |
| | CoT vs ZS | $-2.73$ | $0.006$ | $-0.061$ | 29.9% |
| Qwen 2.5 7B | Few-shot vs ZS | $-0.09$ | $0.929$ | $-0.002$ | 31.4% |
| | CoT vs ZS | $-4.30$ | $< 0.0001$ | $-0.096$ | 29.6% |
| Gemma 2 9B | Few-shot vs ZS | $+4.26$ | $< 0.0001$ | $+0.095$ | 31.2% |
| | CoT vs ZS | $+8.52$ | $< 0.0001$ | $+0.191$ | 36.5% |

Table 15 reports the four-model-average F1 of the baselines under both co-primary metrics (the source of the values quoted in Section 4.1), with zero-shot as reference.

Table 15: Four-model-average F1 for the parametric baselines under both co-primary metrics (main study, symmetric alignment), with zero-shot as reference. Neither baseline exceeds zero-shot at either metric.

| Condition | Type-level F1 | Span-level F1 |
|---|---|---|
| Zero-shot | 0.631 | 0.325 |
| Few-shot | 0.628 | 0.319 |
| Chain-of-thought | 0.606 | 0.298 |

## C.3 Confidence Interval Analysis

Table 16: Bootstrap 95% CI analysis (main study, target margin ±0.05). All cells sufficient at N = 2,000. req_n = minimum n for target margin.

| Model | Condition | n | f1_$\mu$ | std | CI low | CI high | req_n |
|---|---|---|---|---|---|---|---|
| Llama 3.1 8B | with_docs | 2000 | 0.526 | 0.356 | 0.511 | 0.542 | 195 |
| | with_skills | 2000 | 0.504 | 0.258 | 0.493 | 0.515 | 103 |
| | with_tools | 2000 | 0.567 | 0.259 | 0.556 | 0.579 | 104 |
| | zero_shot | 2000 | 0.675 | 0.308 | 0.662 | 0.689 | 146 |
| Mistral 7B | with_docs | 2000 | 0.592 | 0.307 | 0.579 | 0.606 | 145 |
| | with_skills | 2000 | 0.528 | 0.263 | 0.516 | 0.539 | 107 |
| | with_tools | 2000 | 0.528 | 0.263 | 0.517 | 0.540 | 107 |
| | zero_shot | 2000 | 0.584 | 0.309 | 0.570 | 0.597 | 148 |
| Qwen 2.5 7B | with_docs | 2000 | 0.652 | 0.278 | 0.640 | 0.665 | 119 |
| | with_skills | 2000 | 0.484 | 0.273 | 0.472 | 0.496 | 115 |
| | with_tools | 2000 | 0.488 | 0.280 | 0.476 | 0.500 | 121 |
| | zero_shot | 2000 | 0.640 | 0.322 | 0.626 | 0.655 | 160 |
| Gemma 2 9B | with_docs | 2000 | 0.635 | 0.281 | 0.623 | 0.647 | 122 |
| | with_skills | 2000 | 0.526 | 0.252 | 0.515 | 0.537 | 98 |
| | with_tools | 2000 | 0.525 | 0.253 | 0.514 | 0.536 | 99 |
| | zero_shot | 2000 | 0.624 | 0.313 | 0.610 | 0.638 | 151 |

# D Per-PII-Type Recall Tables

Table 17 presents recall by PII type and condition for all four models across the ten most frequent types (main study). Single-turn conditions (ZS, +Docs, few-shot, CoT) and multi-turn (+Tool, +Skills) are shown together; models are grouped within the table.

Table 17: Recall by PII type and model, all conditions and parametric baselines (main study).

| Model | PII Type | ZS | +Docs | +Tool | +Skills | few-shot | CoT |
|---|---|---|---|---|---|---|---|
| Llama 3.1 8B | DATE_TIME | 0.70 | 0.58 | 0.75 | 0.73 | 0.75 | 0.73 |
| | EMAIL_ADDRESS | 0.94 | 0.77 | 0.97 | 0.94 | 0.88 | 0.76 |
| | HEALTH_INSURANCE_ID | 0.73 | 0.17 | 0.08 | 0.01 | 0.52 | 0.36 |
| | IP_ADDRESS | 0.92 | 0.56 | 0.96 | 0.90 | 0.83 | 0.69 |
| | LOCATION | 0.53 | 0.46 | 0.47 | 0.41 | 0.60 | 0.56 |
| | PERSON | 0.65 | 0.47 | 0.48 | 0.40 | 0.63 | 0.67 |
| | PHONE_NUMBER | 0.85 | 0.72 | 0.72 | 0.63 | 0.82 | 0.62 |
| | US_DRIVERS_LICENSE_NUMBER | 0.48 | 0.13 | 0.81 | 0.80 | 0.43 | 0.31 |
| | US_PASSPORT_NUMBER | 0.64 | 0.41 | 0.72 | 0.67 | 0.52 | 0.44 |
| | US_SOCIAL_SECURITY_NUMBER | 0.76 | 0.62 | 0.63 | 0.56 | 0.60 | 0.52 |
| Mistral 7B | DATE_TIME | 0.40 | 0.43 | 0.68 | 0.68 | 0.64 | 0.49 |
| | EMAIL_ADDRESS | 0.93 | 0.90 | 0.97 | 0.97 | 0.91 | 0.90 |
| | HEALTH_INSURANCE_ID | 0.63 | 0.30 | 0.01 | 0.01 | 0.26 | 0.43 |
| | IP_ADDRESS | 0.82 | 0.82 | 0.96 | 0.96 | 0.80 | 0.70 |
| | LOCATION | 0.30 | 0.50 | 0.39 | 0.38 | 0.58 | 0.38 |
| | PERSON | 0.46 | 0.53 | 0.40 | 0.40 | 0.58 | 0.58 |
| | PHONE_NUMBER | 0.85 | 0.87 | 0.62 | 0.62 | 0.85 | 0.78 |
| | US_DRIVERS_LICENSE_NUMBER | 0.46 | 0.34 | 0.69 | 0.68 | 0.34 | 0.39 |
| | US_PASSPORT_NUMBER | 0.57 | 0.59 | 0.54 | 0.54 | 0.58 | 0.52 |
| | US_SOCIAL_SECURITY_NUMBER | 0.79 | 0.70 | 0.52 | 0.52 | 0.69 | 0.69 |
| Qwen 2.5 7B | DATE_TIME | 0.58 | 0.68 | 0.68 | 0.59 | 0.78 | 0.57 |
| | EMAIL_ADDRESS | 0.96 | 0.97 | 0.96 | 0.97 | 0.95 | 0.94 |
| | HEALTH_INSURANCE_ID | 0.74 | 0.15 | 0.02 | 0.00 | 0.55 | 0.46 |
| | IP_ADDRESS | 0.82 | 0.89 | 0.94 | 0.87 | 0.72 | 0.81 |
| | LOCATION | 0.56 | 0.67 | 0.40 | 0.35 | 0.64 | 0.52 |
| | PERSON | 0.54 | 0.51 | 0.38 | 0.35 | 0.63 | 0.63 |
| | PHONE_NUMBER | 0.92 | 0.90 | 0.64 | 0.56 | 0.91 | 0.81 |
| | US_DRIVERS_LICENSE_NUMBER | 0.53 | 0.16 | 0.78 | 0.74 | 0.08 | 0.51 |
| | US_PASSPORT_NUMBER | 0.65 | 0.72 | 0.66 | 0.75 | 0.53 | 0.60 |
| | US_SOCIAL_SECURITY_NUMBER | 0.76 | 0.85 | 0.55 | 0.53 | 0.64 | 0.69 |
| Gemma 2 9B | DATE_TIME | 0.39 | 0.53 | 0.71 | 0.71 | 0.62 | 0.64 |
| | EMAIL_ADDRESS | 0.96 | 0.97 | 0.98 | 0.98 | 0.94 | 0.95 |
| | HEALTH_INSURANCE_ID | 0.72 | 0.10 | 0.01 | 0.01 | 0.64 | 0.55 |
| | IP_ADDRESS | 0.79 | 0.84 | 0.97 | 0.97 | 0.73 | 0.80 |
| | LOCATION | 0.37 | 0.51 | 0.40 | 0.40 | 0.47 | 0.55 |
| | PERSON | 0.62 | 0.62 | 0.42 | 0.42 | 0.60 | 0.69 |
| | PHONE_NUMBER | 0.90 | 0.94 | 0.66 | 0.66 | 0.87 | 0.87 |
| | US_DRIVERS_LICENSE_NUMBER | 0.56 | 0.17 | 0.79 | 0.79 | 0.51 | 0.48 |
| | US_PASSPORT_NUMBER | 0.68 | 0.66 | 0.60 | 0.61 | 0.62 | 0.66 |
| | US_SOCIAL_SECURITY_NUMBER | 0.69 | 0.72 | 0.55 | 0.56 | 0.67 | 0.67 |

## E  Offset-Provenance Tables

Section 4.2 reports four-model averages. Table 18 gives the per-model breakdown on the span-bearing subset, confirming that the collapse in offset generation is uniform and is not driven by any single model: exact-boundary F1 under zero-shot ranges only from 0.0002 (Gemma) to 0.0012 (Qwen), while every detector-backed condition lands between 0.338 and 0.370. Type-level F1 remains in a normal range (0.474–0.644) for all models under all conditions, so the failure is specific to localization.

Table 18: Per-model mean F1 under three matching criteria, span-bearing subset ($n = 1{,}132$, symmetric alignment). Exact-boundary F1 for the parametric conditions is near zero for every model.

| Model | Condition | Type | Span | Strict |
|---|---|---|---|---|
| Gemma 2 9B | Zero-shot | 0.568 | 0.045 | 0.0002 |
| Gemma 2 9B | +Docs | 0.604 | 0.044 | 0.0018 |
| Gemma 2 9B | +Tool | 0.513 | 0.397 | 0.3634 |
| Gemma 2 9B | +Skills | 0.514 | 0.398 | 0.3639 |
| | | | | |
| Llama 3.1 8B | Zero-shot | 0.634 | 0.038 | 0.0003 |
| Llama 3.1 8B | +Docs | 0.474 | 0.028 | 0.0000 |
| Llama 3.1 8B | +Tool | 0.567 | 0.410 | 0.3653 |
| Llama 3.1 8B | +Skills | 0.490 | 0.372 | 0.3384 |
| | | | | |
| Mistral 7B | Zero-shot | 0.527 | 0.035 | 0.0008 |
| Mistral 7B | +Docs | 0.542 | 0.027 | 0.0009 |
| Mistral 7B | +Tool | 0.519 | 0.381 | 0.3489 |
| Mistral 7B | +Skills | 0.518 | 0.382 | 0.3491 |
| | | | | |
| Qwen 2.5 7B | Zero-shot | 0.593 | 0.040 | 0.0012 |
| Qwen 2.5 7B | +Docs | 0.644 | 0.038 | 0.0011 |
| Qwen 2.5 7B | +Tool | 0.480 | 0.382 | 0.3481 |
| Qwen 2.5 7B | +Skills | 0.478 | 0.406 | 0.3697 |

## F  Robustness Control Tables

Tables 19 and 20 give the full underlying numbers for the two robustness controls summarized in Section 5.6: re-running the ablation on a larger (14B) model, and re-running it unquantized.

Table 19: Scale control: Qwen2.5-14B (4-bit) versus the 7–9B four-model average, by condition, over the same 2,000 samples and symmetric alignment. $\Delta$ is 14B minus the 7–9B average. The type-level zero-shot lead over +Tool is +0.085 at 14B (versus +0.104), so the pattern is essentially unchanged one model generation up.

| Condition | Type-level | | | Span-level (IoU $\geq$ 0.5) | | |
|---|---|---|---|---|---|---|
| | 7–9B | 14B | $\Delta$ | 7–9B | 14B | $\Delta$ |
| Zero-shot | 0.631 | 0.605 | −0.026 | 0.325 | 0.322 | −0.003 |
| +Docs | 0.602 | 0.632 | +0.030 | 0.301 | 0.319 | +0.018 |
| +Tool | 0.527 | 0.520 | −0.007 | 0.455 | 0.466 | +0.011 |
| +Skills | 0.511 | 0.521 | +0.010 | 0.448 | 0.467 | +0.019 |

Table 20: Precision control: unquantized (fp16/bf16) versus 4-bit, per model, on the shared stratified 300-sample subset ($n = 1{,}200$ sample-condition pairs per model). $\Delta$ is 16-bit minus 4-bit with its 95% CI. Pooled, the mean difference is $0.006 \pm 0.006$ (type) and $0.004 \pm 0.004$ (span).

| | Type-level | | | Span-level (IoU $\geq 0.5$) | | |
| Model | 4-bit | 16-bit | $\Delta$ | 4-bit | 16-bit | $\Delta$ |
|---|---|---|---|---|---|---|
| Gemma 2 9B | 0.567 | 0.574 | $+0.007 \pm 0.008$ | 0.278 | 0.279 | $+0.001 \pm 0.006$ |
| Llama 3.1 8B | 0.562 | 0.590 | $+0.028 \pm 0.016$ | 0.274 | 0.284 | $+0.010 \pm 0.008$ |
| Mistral 7B | 0.530 | 0.527 | $-0.003 \pm 0.012$ | 0.264 | 0.264 | $0.000 \pm 0.008$ |
| Qwen 2.5 7B | 0.565 | 0.557 | $-0.008 \pm 0.012$ | 0.271 | 0.277 | $+0.007 \pm 0.008$ |

## G  Skill-Viewed Subgroup Analysis

### G.1  Group sizes and mean F1

Table 21: Skill-viewed vs. not-viewed mean F1 (with_skills condition, main study). Llama and Gemma had 100% view rates; no not-viewed group is available for those models. Mean F1 difference is viewed minus not-viewed at the type level.

| Model | Viewed $n$ | Not-viewed $n$ | View rate | Mean F1 difference |
|---|---|---|---|---|
| Mistral 7B | 948 | 1,052 | 47.4% | $+0.011$ |
| Qwen 2.5 7B | 341 | 1,659 | 17.1% | $-0.002$ |
| Llama 3.1 8B | 2,000 | 0 | 100% | n/a |
| Gemma 2 9B | 2,000 | 0 | 100% | n/a |

Analysis shows no discernible difference in mean F1 between viewed and not-viewed groups for Mistral or Qwen. This within-condition comparison controls for all experimental factors except document reading and provides the strongest available evidence that the Skill document itself does not drive performance.

### G.2  Turn count and elapsed time correlations

Table 22: Pearson correlation of turns and elapsed seconds with F1 per sample (main study). n/a = single-turn condition; no turn variance.

| | zero_shot | | with_docs | | with_tools | | with_skills | |
| Model | turns | elapsed | turns | elapsed | turns | elapsed | turns | elapsed |
|---|---|---|---|---|---|---|---|---|
| Gemma 2 9B | n/a | $-0.02$ | n/a | $-0.01$ | n/a | $+\mathbf{0.05}$ | $-0.01$ | $+\mathbf{0.06}$ |
| Llama 3.1 8B | n/a | $-0.16$ | n/a | $-0.04$ | $-0.05$ | $-0.03$ | $-0.11$ | $-0.08$ |
| Mistral 7B | n/a | $-0.16$ | n/a | $-0.08$ | $+0.00$ | $+\mathbf{0.08}$ | $+0.01$ | $+\mathbf{0.07}$ |
| Qwen 2.5 7B | n/a | $-0.07$ | n/a | $+\mathbf{0.02}$ | $+0.02$ | $+\mathbf{0.11}$ | $+\mathbf{0.01}$ | $+\mathbf{0.09}$ |

## H  Skill Document (`SKILL.md`)

The following is the `SKILL.md` document served to models via the `view_skill("pii-detection")` tool call in the with_skills condition. The same document was used for both pilot and main studies. It was compressed to approximately 250 tokens to prevent context overload in 7–9B models (see Section 3).

```
# pii-detection skill

## When to use
When asked to detect, identify, or redact personally identifiable
information (PII).

## Available tool
- 'analyze_pii': Detects PII entities and returns sanitized text.
  Call with: [TOOL_CALL: analyze_pii]

## PII types
The tool returns all PII types supported by PII-Codex (e.g. PERSON,
LOCATION, ADDRESS, DATE, DATE_TIME, PHONE_NUMBER, EMAIL_ADDRESS,
US_SOCIAL_SECURITY_NUMBER, US_PASSPORT_NUMBER, US_DRIVERS_LICENSE_NUMBER,
CREDIT_CARD_NUMBER, IP_ADDRESS, URL, ZIPCODE, and others). Use the tool
output as the source of truth for types and spans.

## Workflow
1. Call [TOOL_CALL: analyze_pii] to get detections and sanitized text
2. Format output as: JSON array of detections, then sanitized text

## Output format
[{"type": "PERSON", "text": "John Smith", "start": 0, "end": 10}]
Then the sanitized text with PII replaced by type labels like
[PERSON], [PHONE_NUMBER].
```

## I  Prompt Templates

The following prompt templates were used for each condition. All models received identical prompts; chat templates were applied per each model's tokenizer specification. {text} is replaced with the sample text at runtime; {docs} is replaced with PII-Codex reference documentation in the with_docs condition.

### I.1  Zero-shot prompt

```
You are a PII (Personally Identifiable Information) detection expert.

## Task

Analyze the text below and identify all instances of PII. For each PII
span, determine its type, the exact text, and the character start and
end positions.

## Output Format

Return your findings as a JSON array only. Each item must have: type
(string), text (exact span), start (character index), end (character
index). If no PII is found, return an empty array.

## Text to Analyze

{text}
```

## I.2 Documentation prompt (+Docs)

```
You are a PII (Personally Identifiable Information) detection expert.

## Background: PII-Codex Library

PII-Codex is a Python library for PII detection, categorization, and
severity assessment. It combines Microsoft Presidio detection with
academic frameworks for risk classification.

## When to Use This Skill

- Analyzing text or datasets for privacy risks before sharing
- Sanitizing user-generated content (social media posts, survey
  responses, etc.)
- Assessing compliance readiness for HIPAA, NIST, or DHS guidelines
- Preparing research data for publication
- Auditing logs or documents for accidental PII exposure

### PII Types Detected

- PERSON (names)
- EMAIL_ADDRESS
- PHONE_NUMBER
- LOCATION (addresses, cities)
- DATE_TIME (birthdates, specific dates)
- US_SSN (Social Security Numbers)
- CREDIT_CARD
- US_PASSPORT
- IP_ADDRESS
- URL
- PASSWORD (API keys, secrets, credentials)

### Risk Level Classification

PII-Codex categorizes detections on a 1-3 scale based on Schwartz &
Solove (2012):

| Level | Definition          | Examples                   |
|-------|---------------------|----------------------------|
| 1     | Non-Identifiable    | URLs, general locations    |
| 2     | Semi-Identifiable   | Partial addresses, age ranges|
| 3     | Identifiable        | Full names, SSNs, email    |

### Compliance Framework Mappings

Each detection is mapped to multiple compliance frameworks:

- NIST SP 800-122: Directly PII, Linked PII, Linkable PII
- HIPAA: Protected Health Information (PHI) identifiers
- DHS: Standalone PII vs Linkable PII
- Milne et al.: Information Sensitivity Typology clusters
```

```
## Best Practices

1. Always analyze before sharing - Run collection analysis on any
   dataset before external sharing
2. Check risk distribution - A high standard deviation indicates mixed
   sensitivity; review manually
3. Use sanitized_text for downstream tasks - The redacted output
   preserves structure while removing PII
4. Review Semi-Identifiable (Level 2) - These may become identifiable
   when combined with other data
5. Log detection frequencies - Patterns in PII types can reveal data
   collection issues

## Task

Analyze the provided text and identify all instances of Personal
Identifiable Information (PII).

For each PII instance found, provide:
1. The PII type (use the types listed above)
2. The exact text that contains the PII
3. The character positions (start and end) where the PII appears

## Output Format

Return your findings as a JSON array:
[{"type": "PII_TYPE", "text": "the actual PII text", "start": 0, "end": 10}]
If no PII is found, return an empty array: []

## Text to Analyze

{text}
```

### I.3    Tool-augmented system message (+Tool)

The with_tools condition uses the LangGraph SkillsAgent with a single available tool (`analyze_pii`). The system message is:

```
You are a PII detection agent with access to the analyze_pii tool.

## Available Tool

**analyze_pii**: Detects Personal Identifiable Information in text
using PII-Codex.
- Returns: JSON with detected PII entities, types, positions, and risk
  levels

To call this tool, respond with EXACTLY:
[TOOL_CALL: analyze_pii]

After receiving tool results, use them to provide your final answer.

## Task
```

```
Analyze the following text for PII. Use the analyze_pii tool to get
accurate detections, then format the results.

## Text to Analyze

{text}

## Instructions

1. First, call the analyze_pii tool by responding with:
   [TOOL_CALL: analyze_pii]
2. After receiving results, format them as a JSON array with type,
   text, start, end for each PII found.
```

### I.4 Skills-augmented system message (+Skills)

The with_skills condition uses the same LangGraph agent with three tools: `list_skills`, `view_skill`, and `analyze_pii`. The system message is:

```
You are a PII detection agent. Detect all PII in the text below.

You have three tools. Call them in order using exact bracket syntax:
1. [TOOL_CALL: list_skills]
2. [TOOL_CALL: view_skill pii-detection]
3. [TOOL_CALL: analyze_pii]

Your first reply must be exactly: [TOOL_CALL: list_skills]

After receiving analyze_pii results, output:
1. JSON array: [{"type": "PII_TYPE", "text": "...", "start": 0, "end": 10}]
2. Sanitized text with placeholders like [PERSON], [PHONE_NUMBER]

If no PII is found, return [].

## Text to Analyze

{text}
```

### I.5 Few-shot prompt

The few-shot baseline prepends three worked in-context examples to the zero-shot prompt (single-turn, no tool execution).

```
You are a PII (Personally Identifiable Information) detection expert.

## Task

Analyze the text below and identify all instances of PII. For each
PII span, determine its type, the exact text, and the character start
and end positions.

## Output Format
```

```
Return your findings as a JSON array only. Each item must have: type
(string), text (exact span), start (character index), end (character
index). If no PII is found, return an empty array.

## Examples

Text: "Contact John Smith at john.smith@acme.com or 555-0142."
Output: [{"type": "PERSON", "text": "John Smith", "start": 8,
  "end": 18}, {"type": "EMAIL_ADDRESS", "text":
  "john.smith@acme.com", "start": 22, "end": 41}, {"type":
  "PHONE_NUMBER", "text": "555-0142", "start": 45, "end": 53}]

Text: "She was born on 1990-03-14 and lives in Austin, Texas."
Output: [{"type": "DATE_TIME", "text": "1990-03-14", "start": 16,
  "end": 26}, {"type": "LOCATION", "text": "Austin, Texas",
  "start": 40, "end": 53}]

Text: "No personal information here."
Output: []

## Text to Analyze

{text}
```

### I.6 Chain-of-thought prompt

The chain-of-thought baseline adds an explicit step-by-step reasoning instruction to the zero-shot prompt (single-turn, no tool execution).

```
You are a PII (Personally Identifiable Information) detection expert.

## Task

Analyze the text below and identify all instances of PII. For each
PII span, determine its type, the exact text, and the character start
and end positions.

## Instructions

First, think step by step: scan the text for names, contact details,
identifiers, dates, locations, and other sensitive spans, noting each
candidate and its character position. Then produce your final answer.

## Output Format

After your reasoning, return your findings as a JSON array. Each item
must have: type (string), text (exact span), start (character index),
end (character index). If no PII is found, return an empty array. The
JSON array must be the final element of your response.

## Text to Analyze

{text}
```

