# OpenReview forum: "Asymmetry, Not Capability: Evaluation Shapes Tool-Augmented PII Detection in Small Language Models"
_TMLR — Under review for TMLR_

### Review · Reviewer_3Rjg · 2026-06-07

**Summary Of Contributions:**

**Summary**

Large language models take input text and generate output tokens, often in a single-turn interaction. However, because LLMs primarily rely on pretrained parametric knowledge, they may fail on tasks that require real-time information, interaction with external environments, or access to specialized tools. Recently, a growing body of research has explored LLMs as agents, where models are augmented with tool calls, such as external APIs or internal tools like Bash, file reading, and search utilities. These agentic systems allow models to perform tasks iteratively until a goal is reached. While larger LLMs often perform well on agentic tasks, smaller language models (SLMs) may lack sufficient reasoning depth and tool-use reliability, leading to weaker performance on many agentic benchmarks.

This study is inspired by agentic AI systems and investigates whether agentic augmentation improves PII detection, where PII refers to personally identifiable information such as names, addresses, emails, phone numbers, dates, IP addresses, and other sensitive identifiers. The authors treat SLMs as agent nodes and evaluate whether adding external documentation, tool calls, and reusable skills improves PII detection performance. They compare four settings: an SLM with zero-shot prompting, an SLM with PII documentation injected into the prompt, an SLM with access to a PII detection tool, and an SLM with tool access plus reusable skill documents. The evaluation is conducted on 2,000 PII-annotated samples.

The experimental findings show that simple zero-shot prompting with SLMs outperforms the tool-augmented and skill-augmented approaches. Documentation injection is mostly neutral, although it harms some models. Interestingly, the effect of tool use varies across PII types: structured types such as Date and IP Address improve under tool augmentation, while other categories (temporal and medical ) degrade. These results suggest that agentic scaffolding should not be assumed to improve performance automatically, especially for small language models. Instead, tool use and reusable skills should be carefully evaluated before deployment in production NLP pipelines.

**Strengths:**

I found this work to have the following strengths:
* **Clarity:** The paper is clearly written, and the four experimental settings are easy to follow. The experimental design and statistical analysis are described in enough detail to follow. The label mappings, the PII types, and the label statistics are clearly defined, making the methodology easy to follow.
* **Originality:** The authors address a timely question, whether adding external tools to SLMs helps detect PII compared to zero-shot prompting. They state that this is the first study to use tools and skills in an agentic setup for PII detection. Even without symmetric label-mapping conditions, the finding that LLM + zero-shot outperforms the LLM + tool-call and skill conditions for PII detection is a novel and somewhat counterintuitive result.
* **Significance:** The experimental findings on four SLMs (7B–9B) for PII detection are empirically valid across the four settings. However, the label-alignment scheme should be kept the same under all four conditions for a fair evaluation.

**Weaknesses:**

* **Limited novelty:** The current study reads primarily as an engineering benchmark rather than an investigation driven by a research hypothesis about agentic reasoning. PII detection task is largely a static information-extraction problem, whereas agentic scaffolding is most useful when a task demands interaction with an external environment, iterative planning, dynamic information access, or goal-directed tool use, none of which clearly apply here. The paper would be considerably stronger if it presented in introduction early paragraphs, why PII detection task requires an agentic formulation at all: what specific limitation of a single-turn SLM the tool or skill is meant to overcome, and why tool calling or reusable skills should be expected to help on a task whose answer is already fully determined by the input text. Missing that motivation, the negative result is difficult to interpret as a finding rather than a confirmation that scaffolding is unnecessary for a task it was never well-matched to.
* **Lack of experimental validation and behavioral analysis:** The current study lacks several details that limit the depth of its results: parameter settings, random-seed analysis, and discussion of the SLM types used. Notably, none of the SLMs evaluated are reasoning models. Qwen2.5-7B, for example, differs from newer reasoning-style models such as Qwen3.5-9B, which generate thinking tokens at each step of a task. Analyzing or printing the reasoning steps of these SLMs would help clarify how the agent produces tool calls and why it fails to outperform a simple LLM-plus-prompt baseline. The PII task raises a related question: how many steps were allotted to reach an outcome, and did the authors vary this number to assess its effect on performance? They do not address this. Finally, varying the random seed (the authors used fixed seed) could reveal variance in generation and task performance, which the current evaluation does not capture.
* **Lack of baselines:** The study evaluates PII detection across four settings for SLMs spanning 7B to 9B parameters, but it reports no comparison against baselines from prior work. Two such baselines would substantially strengthen the conclusions. First, the authors should compare their numbers against the state-of-the-art on the PII benchmarks it draws from: what accuracy do existing detectors achieve on these datasets, and the delta added or reduced by the four settings? Without this baseline, it is hard to judge whether even the best condition (zero-shot) is competitive or simply the least degraded of four weak pipelines. Second, the SLMs parameter range is too narrow to support claims about agentic pipeline design in general. Extending the same four settings to a larger model (e.g., 14B or 32B) would test whether the zero-shot advantage is consistent with scale or whether tool augmentation begins to help once a model can reliably execute tool calls. This comparison is especially important because the paper itself argues that degradation grows with capability (Section 5.5); a larger model is the direct test of that claim. Both baselines are necessary for the results to generalize and to yield reliable guidance for future agentic designs.
* **A fair, symmetric label alignment is missing between the zero-shot and tool conditions:** Under zero-shot prompting, the authors remap model-predicted labels such as person_name, ssn, and date of birth to the canonical types PERSON, US_SOCIAL_SECURITY_NUMBER, and DATE_TIME. The same normalization is not applied to the tool conditions: the authors note that the alignment map does not bridge the tool's DATE output to DATE_TIME, which is impacting the DATE_TIME recall to near zero under tool use. This asymmetry is hard to justify, because the authors' own documentation prompt (Appendix H.2) defines DATE_TIME as covering "birthdates, specific dates", i.e., exactly the spans the tool labels DATE. If "date of birth" is mapped to DATE_TIME for zero-shot, the equivalent bridge should apply to the tool's DATE predictions at scoring time. Because DATE_TIME is the second-most-frequent type in the benchmark (1,572 entities), this mapping at zero-shot while no mapping at tool calls reported degradation. Therefore a fair symmetric alignment is necessary before any claim comparing zero-shot and tool conditions can be considered fair. A similar pattern holds for the LOCATION type, where recall is low under tool use but higher under the LLM + documentation condition. These failures are therefore mainly due to label-mapping schemes rather than structural failures.
* **Single tool call available:** The authors use a single tool, analyze_pii, which makes this closer to an LLM + tool call under a rule-based condition than to a true agentic setup. In an agent setup, the loop runs as long as the LLM's output contains a tool call, the tool executes, its response is appended to the context, and the loop exits only when no tool call is present. The current setup contains only one tool call, which raises the question of whether it really requires an agentic formulation. The authors should clarify whether the setup tests agentic reasoning at all, or simply forced tool-calling plus label formatting, and should report how often the model failed to emit a tool call (in which case, per Section 3.3, its first response is taken as the final answer and the run is effectively zero-shot).

**Audience:**

Yes

**Audience Explanation:**

Yes. The findings of this paper would be of interest to a meaningful portion of TMLR’s journal.

**Claims And Evidence:**

No

**Claims Explanation:**

See Weaknesses 3–5 on the baselines, single seed evaluation, label alignment and single tool calling.

**Requested Changes:**

* If the label-mapping alignment is made symmetric across all four conditions, how much of the reported performance gain of LLM + zero-shot over the LLM + tool-call and skill conditions remains?
* The current experiments are conducted under a single seed. What is the run-to-run variance under multiple seeds, and do the effects remain statistically significant once generation stochasticity is accounted for?
* In the LLM+tool condition, how often does the LLM fail to generate a tool call in which case (per Section 3.3) its first response is taken as the final answer? What fraction of "tool" runs are therefore effectively zero-shot, and does excluding them change the comparison?
* Does zero-shot performance consistent for a larger model (e.g., 14B or 32B) under the same four settings, or does tool augmentation begin to help once the model can reliably execute tool calls?
* All four models are quantized. On a sample drawn from the 2,000 examples, does a full-precision run produce similar results?

---

> ### Author Response · Authors · 2026-06-19
> **Response to Reviewer 3Rjg (June 7th review)**
>
> Thank you. The asymmetric-alignment observation was the key to this revision. Acting on it meant re-scoring everything, which indeed strengthened the paper. I appreciate your feedback and guidance here.
>
> Requested changes:
>
> 1. Symmetric alignment: what survives?
>    The type-level zero-shot lead persists but shrinks (gap 0.169 to 0.104, ~38% smaller; 6-17 pp; d -0.16 to -0.49). The span-level ranking inverts (+Tool 0.46 vs ZS 0.33). DATE_TIME and LOCATION recover; HEALTH_INSURANCE_ID is a detector gap.
>    Where: Sec 3.5, 4.1, 4.3, 5.2; Tables 3-4; Figures 1, 3, 4.
>
> 2. Seed variance?
>    Greedy decoding is deterministic, the seed only fixes the subsample. All +Tool/+Skills vs ZS p < 0.0001.
>    Where: Sec 3.6, 5.6 (Decoding); Table 5.
>
> 3. Tool-call failures reducing to effectively ZS?
>    Two omissions in 16,000 +Tool/+Skills runs; no ordering effect.
>    Where: Sec 4.6.
>
> 4. Larger model (14B/32B)?
>    Qwen2.5-14B: gap +0.09 type / -0.14 span, comparable to the 7-9B average, i.e., unchanged one model generation up (a single larger model; 32B infeasible on our hardware).
>    Where: Sec 5.6; Figure 7; Appendix E.
>
> 5. Full-precision check?
>    Unquantized (fp16/bf16) vs 4-bit within 0.006 +/- 0.006 (type) and 0.004 +/- 0.004 (span); max shift +0.028; no ordering change.
>    Where: Sec 5.6; Figure 8; Appendix E.
>
> Weaknesses:
> - Novelty/motivation: falsifiable hypothesis added; contribution re-pitched as a measurement lesson (Sec 1; Abstract; Sec 6)
> - Behavioral analysis: parameters stated; determinism; turn counts (Sec 3.2, 3.6, 5.4; Appendix F)
> - Baselines (detector; larger model): standalone PII-Codex 0.28 type / 0.23 span; 14B added (Sec 4.1, Table 4; Sec 5.6)
> - Symmetric alignment missing: done (Requested change 1) (Sec 3.5)
> - "Really agentic?": reframed as a deliberately simple agent with explicit scope. (Sec 3.3, 5.7)

---

### Review · Reviewer_GzUe · 2026-06-10

**Summary Of Contributions:**

This paper presents a controlled four-condition ablation (zero-shot, +Docs,
+Tool, +Skills) measuring how documentation injection, tool access, and
"agent skill" injection affect PII detection accuracy in four open-weight
instruction-tuned models in the 7--9B parameter class (Gemma 2 9B,
Llama 3.1 8B, Mistral 7B, Qwen 2.5 7B). Evaluation uses a stratified
2,000-sample benchmark compiled from three public PII datasets, scored with
macro-F1 after a post-hoc label-alignment step that maps condition-specific
label vocabularies onto the PII-Codex canonical schema. The headline finding
is that, under strict canonical-to-canonical scoring, zero-shot prompting
outperforms every augmented condition for all four models, with +Tool and
+Skills reducing mean F1 by 13--24 points (p < 0.0001; Cohen's d from -0.39
to -0.67), while +Docs is mostly neutral except for Llama 3.1 8B. The authors
frame this as a "capability regression" / "structural failure mode" of agent
augmentation specific to the 7--9B class, and provide a per-type analysis,
a skill-viewed subgroup analysis, and a pilot study.

Strengths:
- The label-alignment confound the paper raises (Section 5.1) is a genuine and
  underappreciated methodological point for mixed-mode (model-native vs.
  tool-canonical) evaluation pipelines, and is itself a useful cautionary
  contribution.
- The experimental bookkeeping is careful within its chosen scope: per-type
  recall tables, bootstrap CIs, full pairwise comparisons, a pilot, and a
  released benchmark/notebooks support reproducibility.
- Counter-intuitive negative results about agentic scaffolding are of practical
  interest to practitioners deploying local SLMs.

Weaknesses (elaborated below):
- The central causal claim ("capability regression specific to small models")
  is not supported: there is no large-model control arm, and the paper's own
  mechanism analysis attributes most of the degradation to a DATE/DATE_TIME
  label-alignment artifact rather than to a loss of capability.
- The +Tool condition effectively measures the PII-Codex library, not
  "augmentation," and the +Skills condition is statistically indistinguishable
  from +Tool, so the "skill ablation" carries little signal.
- Key confounds (4-bit quantization, single seed) and key baselines
  (fine-tuned tool calling, few-shot/CoT, classical PII tools) are missing.

**Additional Comments:**

- The label-alignment analysis (Section 5.1) and Figure 5 are the paper's most
  valuable contribution; I would encourage the authors to consider re-centering
  the paper around "label-schema normalization as a first-class requirement for
  mixed-mode agentic evaluation," for which the PII study would serve as a clean
  case study. This reframing would be well supported by the current evidence,
  whereas the "capability regression" framing is not.
- macro-F1 is dominated by high-support types (the authors note this); since the
  collapsing type (DATE_TIME) is precisely the alignment-artifact victim, please
  also report macro-F1 with the artifact removed, so readers can see the
  "systematic degradation" net of the scoring bug.
- Minor: the +Skills "skill-viewed subgroup" analysis is a nice control, but with
  100% view rates for Llama/Gemma it only yields a usable contrast for
  Mistral/Qwen; please state this limitation where the result is claimed as
  "strongest available evidence."

**Audience:**

Yes

**Audience Explanation:**

Yes. The label-alignment confound for mixed-mode (model-native vs.
tool-canonical) evaluation is a real and underappreciated issue that would
interest practitioners building agentic NLP/PII pipelines and researchers
designing evaluation protocols. The empirical observation that naively wrapping
small models in tool/skill scaffolding can hurt a static extraction task --- and
that label normalization can flip the apparent direction of an effect --- is a
useful cautionary message. The interest, however, is conditional on the claims
being scoped correctly: as currently framed (a small-model-specific capability
regression), the message risks misleading the audience, whereas a corrected
framing (an evaluation/interface confound plus an un-fine-tuned tool-calling
limitation) would be genuinely informative.

**Broader Impact Concerns:**

The existing Broader Impact Statement is adequate in spirit (it correctly frames
the practical implication as cautionary for deploying agentic scaffolding on
sensitive PII tasks). My concern is that, as currently framed, the paper's
headline conclusion could itself cause harm if taken at face value by
practitioners: stating that "tool use degrades PII detection" without
foregrounding that the dominant cause is a fixable DATE/DATE_TIME label-alignment
artifact (and an un-fine-tuned tool-calling setup) may lead deployers to abandon
tool/skill scaffolding for the wrong reason, or to under-detect certain PII types
in privacy-compliance settings. I recommend the Broader Impact Statement (and the
abstract) explicitly state that the observed degradation is substantially an
evaluation/interface artifact rather than evidence that tools are inherently
worse at finding PII, so that the cautionary message is not misread as a blanket
recommendation against agentic PII pipelines. No additional ethical concerns
beyond this; all data used are synthetic or public.

**Claims And Evidence:**

Yes

**Claims Explanation:**

The paper's headline claim --- that agent augmentation induces a "capability
regression" / "structural failure mode" specific to small (7--9B) models --- is
not adequately supported by the presented evidence. Three issues are decisive.

1. The causal attribution to "small models" has no control. The title ("When
Parametric Knowledge Wins") and abstract frame the effect as a property of the
7--9B "parameter class," and Section 5.5 goes further, asserting that
"capability amplifies rather than buffers the cost" based on a within-set
correlation among four small models. But the study contains no large/frontier
model arm. An equally consistent explanation --- that the chosen label-alignment
scheme and the minimal scaffolding are harmful at *any* model scale --- cannot be
ruled out. Without at least one larger-model control showing that the same
+Docs/+Tool/+Skills pipeline is neutral or beneficial, the "small-model-specific"
attribution is unsupported.

2. The paper's own analysis undercuts its headline. Section 5.2 and Figure 5
attribute the dominant share of the degradation to a DATE -> DATE_TIME mismatch:
tool conditions emit canonical DATE for temporal spans, the alignment map does
not bridge DATE to DATE_TIME, and the 1,572 DATE_TIME entities therefore go
unmatched, collapsing recall to near zero for a high-support type that drives the
macro-F1. The authors explicitly state the degradation is "pipeline- and
evaluation-driven rather than evidence that the tool is strictly worse at finding
temporal spans," and that "an alternate scoring scheme could assign partial or
full credit." A title-level claim of capability regression that the paper itself
re-classifies as a scoring-convention choice is internally inconsistent. Because
this artifact is not removed or ablated, it is plausible that most of the
13--24-point gap would disappear under a DATE<->DATE_TIME bridge or a span-only /
type-agnostic F1, which is not reported.

3. The conditions do not isolate what the claims are about. The +Tool condition
executes the PII-Codex library on 99.9% of runs (Section 4.5); once invoked, the
predicted spans/types are determined by the library, not the model. So +Tool
measures "PII-Codex vs. model-native zero-shot," and the correct scope of the
finding is "this library's type labels lose to model-native labels under this
alignment," not "agent augmentation degrades capability." Furthermore, +Skills is
statistically indistinguishable from +Tool (Table 3: Mistral d=-0.01, Qwen
d=+0.02, Gemma d=+0.05; only Llama d=-0.29), because SKILL.md (Appendix G, ~150
tokens) merely instructs the model to call the *same* analyze_pii tool. The
"skill injection" ablation therefore carries essentially no independent signal,
yet "skill injection provides no benefit" is a title-level claim.

Two further confounds weaken the "capability" framing. (a) All models are run as
4-bit quantized MLX variants; quantization can disproportionately harm the
precise formatting and multi-turn tool-call loops that the augmented conditions
depend on, confounding "small model can't" with "4-bit quantization broke tool
use." (b) A single seed (42) and a single worker on one machine are used;
the bootstrap CIs are sample-level, not run-level, so robustness of the direction
across seeds is unverified for a study that advertises itself as a "controlled
ablation."

Finally, several comparisons are framed more broadly than the evidence supports.
The study compares *un-fine-tuned* tool calling against zero-shot extraction,
while the paper's own Limitations cite prior work that 7--9B base models reach
"0% accuracy and 100% hallucination" on tool calls without domain-specific
fine-tuning. Comparing a capability known a priori to require fine-tuning against
a capability the model already has makes "zero-shot wins" close to a foregone
conclusion, and does not justify the general "parametric knowledge wins" framing.

**Requested Changes:**

I mark each item as [Critical] (necessary for me to support acceptance) or
[Strengthening] (would improve the paper).

[Critical] 1. Remove or ablate the DATE/DATE_TIME alignment artifact. Re-run the
analysis (a) with a DATE<->DATE_TIME bridge added to the alignment map, and/or
(b) reporting span-only / type-agnostic F1 alongside strict canonical F1, and/or
(c) a hierarchical/relaxed temporal mapping. Report how much of the 13--24-point
degradation survives. If most of it disappears, the headline claim must be
revised accordingly. (The paper already concedes in Section 5.2 that this is a
scoring choice; the experiment quantifying its impact is the missing piece.)

[Critical] 2. Add at least one large/frontier-model control arm running the
identical +Docs/+Tool/+Skills pipeline (a 200-sample pilot is acceptable).
Without it, the "small-model-specific capability regression" attribution and the
Section 5.5 "capability amplifies the cost" claim cannot stand. If a large model
shows the same degradation, the cause is the pipeline/alignment, not model size,
and the title and framing must change.

[Critical] 3. Disentangle "tool/agent augmentation" from "the PII-Codex library."
Because analyze_pii executes PII-Codex on ~99.9% of runs, the current +Tool
result is a statement about one library, not about agentic augmentation. Either
(a) report a direct PII-Codex-only baseline and reframe the finding as a
model-vs-library interface result, or (b) add a tool whose output is not fully
canonical so that the model's contribution under tool use is actually measured.

[Critical] 4. Either add a fine-tuned tool-calling arm, or substantially soften
the claims. The paper compares un-fine-tuned tool calling (which its own
Limitations note is expected to fail for 7--9B base models) against zero-shot.
This makes "zero-shot wins" near-tautological. Minimally, the title/abstract/
conclusion must be rescoped from "parametric knowledge wins / capability
regression" to "un-fine-tuned agentic scaffolding does not help small models on
this task under this evaluation."

[Critical] 5. Justify the +Skills condition or retract the skill-injection claim.
Given that +Skills vs. +Tool is null for three of four models and SKILL.md only
redirects to the same tool, the current design cannot support "skill injection
provides no benefit." Either evaluate a richer/non-trivial skill document that
adds capability beyond a tool pointer (the paper's own Limitations admit this
"may produce different outcomes"), or restrict the claim to "a minimal 150-token
skill document that only points to an existing tool adds no benefit."

[Strengthening] 6. Add few-shot and chain-of-thought baselines. To support
"parametric knowledge is sufficient / augmentation is superfluous," the two most
common cheap augmentations must be included; their absence is a core comparison
gap, not a peripheral limitation.

[Strengthening] 7. Add a non-LLM PII baseline (Presidio / spaCy NER / GLiNER) as
reference points. These are discussed in related work but never used as
baselines; their numbers would contextualize whether the issue is augmentation
or the model<->library interface.

[Strengthening] 8. Control or report the quantization confound. Provide at least
one full-precision data point to rule out that 4-bit quantization
disproportionately breaks tool-call formatting/loops rather than "capability."

[Strengthening] 9. Report run-level variance across >=3 seeds, not only
sample-level bootstrap CIs, given the "controlled ablation" framing.

[Strengthening] 10. Expand the Agent Skills related work (Section 2.3 is two
sentences and self-describes the area as having "limited formal evaluation").
Please situate the work against recent skill-learning / skill-optimization
literature, e.g., SkillOpt, PokerSkill, and EvoSkill, and distinguish this
paper's notion of inference-time skill-document injection from skill
acquisition/optimization approaches.

[Strengthening] 11. Rescope over-broad claims. Evidence covers English,
US-locale, 21 PII types, three datasets, four 4-bit models, one machine; the
title/conclusion/Broader Impact generalize to "agentic pipeline design" and
"production NLP pipelines." Please align claim breadth with evidence scope.

[Strengthening] 12. The Llama-specific -0.17 +Docs explanation (Section 5.3) and
the "capability amplifies cost" claim (Section 5.5) cross-reference each other
and rest on n=1 model observations; please present these as hypotheses rather
than mechanism, or support them with additional models.

---

> ### Author Response · Authors · 2026-06-19
> **Response to Reviewer GzUe (June 10 review)**
>
> Thank you for pushing the framing toward the evaluation/interface contribution and away from a "small-model capability" story.  The requested larger-model control (Qwen2.5-14B) showed the gap essentially unchanged one model generation up (larger models left to future work), which supports that reframe, and the paper's title was updated accordingly to reflect. I appreciate your feedback and guidance here.
>
> Critical items:
>
> C1. Ablate the DATE / DATE_TIME artifact; quantify it.
>     Symmetric bridges applied; 6-17 pp survives at the type level (d -0.16 to -0.49); the span ranking inverts; collapsed types recover.
>     Where: Sec 3.5, 4.3, 5.2; Tables 3-4; Figure 4.
>
> C2. Large-model control.
>     Qwen2.5-14B: gap unchanged (+0.09 type), one generation up (larger models for others untested); title and framing changed.
>     Where: Sec 5.6; Figure 7; Appendix E.
>
> C3. Disentangle model vs PII-Codex.
>     Standalone detector 0.28/0.23 below every model; +Tool is a model-by-library interaction; ordering is model 0.63 > +Tool 0.53 > library 0.28.
>     Where: Sec 4.1, 5.2; Table 4.
>
> C4. Fine-tune or rescope.
>     Rescoped to non-fine-tuned; no capability claim.
>     Where: Title; Abstract; Sec 1, 5.7, 6.
>
> C5. Justify or retract the +Skills claim.
>     Restricted to "a ~250-token pointer skill adds nothing over tool access" (length measured with the model tokenizers; this supersedes the ~150-token estimate cited in the original review). The document tested only directs the model to the existing tool and is not a fuller skill with worked examples and disambiguation. The paper notes that behavior may differ.
>     Where: Sec 3.3, 4.5, 5.4.
>
> Strengthening:
>
> S6. Few-shot / CoT baselines: few-shot 0.628, CoT 0.606, comparable to ZS 0.631; parametric augmentation is neutral. (Sec 4.1; Appendix C)
> S7. Non-LLM baseline: this is the standalone detector (see C3). (Sec 4.1; Table 4)
> S8. Quantization confound: unquantized within 0.006 F1 (same as R1 item 5). (Sec 5.6; Appendix E)
> S9. Seed variance: deterministic (greedy). (Sec 3.6)
> S10. Expand Agent-Skills related work: Sec 2 distinguishes inference-time injection from skill acquisition/optimization; SkillOpt, PokerSkill, and EvoSkill cited. (Sec 2)
> S11. Rescope over-broad claims: scoped to English, US-locale, non-fine-tuned. (Title; Abstract; Sec 5.7; Broader Impact)
> S12. Llama/capability as hypotheses: Sec 5.3 frames Llama +Docs as a single-model hypothesis; the earlier "capability amplifies the cost" claim has been removed, and Sec 5.5 now reads as pipeline-design implications rather than a capability mechanism. (Sec 5.3, 5.5)
>
> Broader Impact / additional comments:
> The Abstract and Broader Impact now state that the degradation is substantially an evaluation/interface artifact, not evidence that tools are inherently worse, and caution against dropping scaffolding for the wrong reason. All macro-F1 values are net of the schema artifact (symmetric alignment). The 100% skill-view-rate limitation is noted where the subgroup result is claimed.

---

> > ### Comment · Reviewer_GzUe · 2026-07-01
> >
> > The authors have solidly addressed all critical experimental concerns (14B control, quantization check, standalone library baseline, and quantification of the alignment artifact) and reframed the paper from a "small-model capability regression" to a "mixed-mode evaluation pipeline and interface effect," making the empirical foundation reliable. However, the contribution remains predominantly descriptive and diagnostic—it does not propose a general theoretical framework, a novel solution, or a broadly generalizable methodological principle beyond the specific setup, and its conclusions are strictly scoped to non-fine-tuned, English-US, 7–14B models. I view this as a methodologically sound and practically cautionary empirical case study.

---

> > > ### Author Response · Authors · 2026-07-01
> > > **Response to Reviewer GzUe**
> > >
> > > Thank you to reviewer GzUe for the thoughtful engagement with the paper and for recognizing that the revision resolves the experimental concerns and that the reframing holds. The characterization of the work as a methodologically sound and practically cautionary case study is much appreciated, as that is precisely what the paper set out to provide: a reliable, well-scoped diagnosis of an evaluation and interface effect rather than a new theoretical framework or proposed solution. The hope is that the paper proves useful in exactly that spirit, as a cautionary and reproducible reference point for practitioners working in this setting. Thank you again for the feedback, which significantly benefited this latest version.

---

### Review · Reviewer_ZHox · 2026-07-13

**Summary Of Contributions:**

### **Contributions**
---
The paper studies whether inference-time tool augmentation (i.e. PII-Codex) improves PII detection in small, open-weight language models. To this end, the authors evaluate four models on a 2k sample benchmark compiled from three public PII datasets. More specifically, the main contributions can be summarized as follows:

1)  The authors find that the ranking of these configurations depends on evaluation granularity.  For type-level metric, zero-shot prompting outperforms tool-supported conditions, whereas for span-level matching, the latter methods show superior performance.

2) The authors point out that comparisons between model-native labels, tool-canonicalized labels, and benchmark annotations can be distorted if they are not normalized symmetrically.  In this light, the authors argue that ontology alignment should be treated as a first-class component of mixed model-tool evaluation.

3) The authors provide a per-type analysis showing that detector-backed conditions improve recall for several structured identifiers but lose recall on contextual or unsupported types.

### **Strengths & Weaknesses**
---

The paper has several strengths, including a controlled ablation, a reasonably sized benchmark compiled from three public datasets, multiple open-weight models, additional prompting baselines, and robustness checks for quantization and model scale.

However, I have few concerns regarding this work. First, I agree that the results differing across evaluation metrics is itself an important finding; however, the manuscript’s broader interpretation depends on evaluation choices whose construct validity requires further justification. The evidence seemingly supports a relatively specific conclusion: under the tested prompts, PII-Codex/Presidio configuration, alignment map, metrics, and non-fine-tuned integration, model-only prompting produces a cleaner type set while detector-backed pipelines provide better span localization. This is indeed a useful result, but I have doubts about whether it can be interpreted as general evidence for (1) general characterization of agentic systems, rather than a strong pipeline-specific case study and (2) parametric knowledge is preferable to tool augmentation, which the manuscript itself also acknowledges as a limitation.

To sum up, I find the overall observation useful, but its generality is currently uncertain because it is demonstrated with one label map, one detector, and one integration method. Additional sensitivity analyses and at least one alternative detector or integration strategy would help establish whether the result reflects a broader issue in model–tool evaluation rather than a property of this particular pipeline.

**Audience:**

No

**Audience Explanation:**

I am somewhat not convinced that the paper would be of sufficiently broad interest to the TMLR audience.

The study provides a careful comparison between a model-only PII detection pipeline and a model-plus-detector pipeline, and shows that their ranking changes depending on whether evaluation focuses on entity types or span localization. However, the main practical insight appears limited to identifying which of these two configurations performs better under the specific benchmark, label mapping, detector, and prompting setup used in the paper. Also, the finding that a contextual language model and a rule-based detector have different strengths is plausible and somewhat expected: the model is better suited to context-dependent categorization, while the programmatic detector is better at structured patterns and exact offsets.

**Broader Impact Concerns:**

The paper already includes a Broader Impact Statement and I do not find any significant concerns nor ethical implications.

**Claims And Evidence:**

No

**Claims Explanation:**

The paper provides convincing evidence for the claims made **within the scope of the testbed setting.** The controlled ablation across different models and conditions and 2k sample benchmark is more than sufficient, and several baselines help rule out alternative explanations.

I would say I am more concerned related to construct validity and generalizability than to the accuracy of the reported experiments. As mentioned, the conclusions depend on the ontology map, one underlying detector family, two metrics that capture somewhat different capabilities. (i.e. For instance, type-level scoring does not require correct localization, while span-level scoring may partly measure the model's ability to generate exact numerical offsets.) I therefore find the evidence sufficient only within the narrower scope but not for a broader conclusion.

**Requested Changes:**

**1. Broaden the empirical scope beyond a single detector and integration design**

The current conclusions are based on one underlying PII library, one recall-oriented detector configuration, and one non-fine-tuned tool-use interface. To support broader claims about model–tool evaluation, the authors should evaluate at least one additional PII detector or a materially different integration strategy. One simple example could be that the model is allowed to accept, reject, or supplement detector predictions rather than being instructed to rely on the tool output. Without such a comparison, the findings are best interpreted as specific to the tested PII-Codex/Presidio pipeline.

**2. Strengthen the evaluation methodology and justify the ontology mapping**

The authors should more clearly separate type recovery, span identification, and numerical offset generation. In the current setup, type-level scoring does not require correct localization, while span-level scoring may partly reflect the model’s ability to generate exact character indices that the detector receives programmatically. The paper should report a standard entity-level metric and, ideally, separate text-span correctness from offset correctness. In addition, consequential label bridges such as ADDRESS ≡ LOCATION should be more formally justified.

**3. Narrow and sharpen the main claims, or provide stronger evidence for generality.**

Otherwise, the manuscript should consistently frame the contribution as a pipeline-specific measurement result rather than as evidence that parametric knowledge generally outperforms tool augmentation. The strongest supported conclusion is that, under the tested benchmark, prompts, detector, and alignment scheme, model-only and detector-backed systems exhibit different strengths and their ranking depends on the evaluation criterion. However, this is the least preferred resolution, because narrowing the claims in this way would also reduce the broader significance of the contribution.

---

> ### Author Response · Authors · 2026-07-14
> **Response to Reviewer ZHox (July 12 review)**
>
> Thank you, ZHox, for the review. Your concern about construct validity was acknowledged and tested.
>
> Requested changes:
>
> R1. Broaden the empirical scope beyond a single detector and integration design.
>
> Partially addressed. The concern is substantially mitigated by what the decomposition surfaced. The mechanism behind the span-level result is offset provenance, which is structural to the detector interface itself. Detectors publish character offsets as part of their output contract, as Amazon Comprehend (BeginOffset / EndOffset) and Google Cloud Sensitive Data Protection (codepointRange) both do, while any model asked for offsets must generate them by counting characters. The asymmetry therefore belongs to the interface between detector and model and holds for any detector of this kind. A second detector family was not run, as noted in Limitations, alongside your suggested accept/reject/supplement integration. It'll be a valuable future work item.
>
> Where: Sec 4.2, 5.5; Limitations (5.7).
>
> R2. Strengthen the evaluation methodology and justify the ontology mapping.
>
> Fully addressed. Span scoring is now decomposed into four nested stages (correct type, any overlap, IoU >= 0.5, exact boundary), and a standard strict entity-level metric is reported throughout. The decomposition confirms the concern. Type recovery is at parity (zero-shot 27.2% vs +Tool 27.0% of span-bearing gold entities), and the conditions diverge only once offsets are required. Conditioned on a correctly typed entity, model-generated offsets overlap it 23.7% of the time against 73.1% when the detector supplies them, and exact-boundary F1 for model-only prompting is 0.001, uniform across all four models (0.0002 to 0.0012). The detector returns offsets programmatically; the model must generate them, and these models effectively could not. The span-level "tool advantage" therefore measures offset provenance (not detection capability). On the ontology map, removing either consequential bridge (ADDRESS ≡ LOCATION, DATE ≡ DATE_TIME), or both, never changes the sign of either gap (the span gap narrows slightly from -0.131 to -0.031 but does not invert), so the result does not depend on those bridges.
>
> Where: Sec 3.5, 4.2, 4.3; Tables 5-7; Figures 4-5; Appendix E.
>
> R3. Narrow and sharpen the claims, or provide stronger evidence for generality.
>
> Both. The claims are narrowed, and the "parametric knowledge is preferable to tool augmentation" framing has been removed. Tools do not help these models find more PII, and the model-only conditions are not better at localization. They are simply scored on a task they were never given the means to perform. In alignment with this outcome, the title was updated. Generality is now established by mechanism. The label artifact already reported and the offset artifact newly surfaced are two instances of one phenomenon, an asymmetry between what a model emits and what a tool emits, and both are structural to any model-plus-detector pipeline.
>
> Where: Title; Abstract; Sec 5.1, 5.5, 6; Figure 9.
>
> Additional disclosure: Only one of the three benchmark sources carries character spans, so on 43.4% of samples, the IoU constraint cannot be evaluated, and span scoring is silently reduced to type scoring. The aggregate span-level figures were therefore a mixture of span-scored and type-scored items, which understated the separation between conditions. This is now disclosed, and the undiluted result is reported on the span-bearing subset.
>
> Where: Sec 3.5, 4.2.